# Saharan warm air intrusions in the Western Mediterranean: identification, impacts on temperature extremes and large-scale mechanisms

Pep Cos[1,2], Matias Olmo[1], Diego Campos[1,2], Raül Marcos-Matamoros[2], Lluis Palma[1,2], Ángel G. Muñoz[1], and Francisco J. Doblas-Reyes[1,3]

[1]Earth Sciences Department, Barcelona Supercomputing Center (BSC), Barcelona, Spain
[2]Department of Applied Physics, University of Barcelona, Barcelona, Spain
[3]Institució Catalana de Recerca i Estudis Avançats (ICREA), Barcelona, Spain

**Correspondence:** Pep Cos (josep.cos@bsc.es)

**Abstract.** Saharan warm intrusions are air masses that develop over the Sahara Desert and can be advected into surrounding areas, creating anomalous atmospheric conditions in those regions. This paper focuses on the characteristics of these intrusions into the Western Mediterranean region (WMed) and their relationship with extreme temperatures in the neighbouring areas during the recent past (1959-2022). We describe and evaluate a methodology to identify Saharan air masses throughout the year and, consequently, a historical catalogue of intrusion events that reach the WMed is built. To identify which large-scale phenomena might be relevant for the formation of the intrusions, we first identify different intrusion types (IT) through a clustering procedure. Different ITs are found for the four seasons, which discriminate the intrusions according to their longitudinal position over the Mediterranean region and their intensity. Upper-tropospheric anomalies are linked to the onset of these events, in particular, an anomalous geopotential high over the intrusions region that slows down the upper-tropospheric circulation over northern Africa. These events are very relevant as they impact extreme temperatures throughout the year and account for a high percentage of the extreme temperature events recorded in the WMed and neighbouring regions in summer.

## 1 Introduction

Numerous studies discussing the Mediterranean climate have underscored important historical and projected changes in temperature, precipitation, and their extremes (Lionello and Scarascia, 2018; Giorgi, 2006; Michaelides et al., 2018; Cos et al., 2022; Olmo et al., 2024), with significant impacts on ecosystems, economies, and humans (Seager et al., 2019; Cherif et al., 2020; Cramer et al., 2018). In the context of climate change, it is crucial to develop robust methods for estimating the future state of the Mediterranean climate. Confidence in model projections depends on, among other things, our understanding of the physical mechanisms driving the observed climate (Barriopedro et al., 2023). To interpret both observational data and climate model simulations effectively, researchers have worked to unravel the physical processes that shape climate patterns. In the case of the Mediterranean some studies have focused on the drivers behind the mean state of temperature and precip-

itation (Tuel and Eltahir, 2020; Brogli et al., 2019; Bladé et al., 2012b), while others have explored the mechanisms behind climatic-impact drivers like heatwaves and extremes, analysing both large-scale circulation and interaction of climate modes of variability (Alvarez-Castro et al., 2018; Faranda et al., 2023; Horton et al., 2015; Muñoz et al., 2015, 2016, 2017) and local processes (Materia et al., 2022; Vogel et al., 2017; Urdiales-Flores et al., 2023; Nabat et al., 2015). Yet, some climate processes and phenomena require further investigation to improve our ability to predict, communicate, and adapt to present and future changes in the Mediterranean climate. Among those phenomena, intrusions of warm air masses from the Sahara that get advected northward (Sousa et al., 2019) have received particularly scarce attention.

There is a large body of literature about Saharan dust intrusions in the Mediterranean, which is very relevant for air quality and human health. However, the climatological properties of the intrusions of warm air, which might be accompanied, or not, by Saharan dust, are not well known. And this is in spite of indications of authors like Sousa et al. (2019), who suggested that warm air intrusions from the Sahara can be a driver of heat waves in the Iberian Peninsula. In the broader Euro-Mediterranean region several studies have used different approaches to better understand the provenance and evolution of air masses that end up producing extreme temperatures at certain locations, through case studies (Rousi et al., 2023; Hotz et al., 2024; Mayer and Wirth, 2025) and analysing extreme temperature events catalogues (Santos et al., 2015; Zschenderlein et al., 2019). While some of the results hint that there is air advected from the Sahara in some of the events, there is no specific study that attempts at generalising the effects of Saharan warm air intrusions. Some work exists for summer intrusions (Sousa et al., 2019; Galvin, 2016), although here we aim to generalise the definition for the whole year. Further work is required to obtain a wider picture of what a Saharan warm air intrusion is and what its implications on the broader Western Mediterranean temperature extremes are. We aim at widening the scope of the Saharan intrusions studies by focusing primarily on temperature, rather than on the far more studied dust storm and transport phenomenon (Cuevas-Agulló et al., 2023).

Several works have hinted at the impact that southerly flow from Africa might play a role in Euro-Mediterranean temperatures. Sousa et al. (2019) studied the impact that past intrusion events have on heat waves in the Iberian Peninsula, Pereira et al. (2005) demonstrated the effect that advections from northern Africa can have on forest fires in Portugal, Santos et al. (2015) suggested that an anticyclonic circulation over the Northwestern Africa generates extremely high temperatures in the Iberian Peninsula, and Zschenderlein et al. (2019) show how air masses in Africa can be linked to extreme temperatures in Central Mediterranean; among others (Sousa et al., 2018; Rousi et al., 2023). These studies made us wonder what the impact and contribution, specifically of Saharan warm air intrusions, are in generating extreme temperature days and what the spatial extent of its area of impact is. Besides, from a large-scale perspective the question of what synoptic configurations lead specifically to Saharan warm air intrusions emerges. We think that, apart from describing their climatological characteristics, it is important to link the Saharan warm air intrusions both to the impact they have on temperature and to the larger scale circulation that is associated with their onset.

In order to better understand the phenomenon and its processes from a climatological point of view, an objective identification algorithm for these masses is required. The algorithm would allow us to build a historical catalogue of events that can be used to gain insight into the phenomenon, its impacts and characteristics. Moreover, this identification and assessment framework should be easily applicable to climate model simulations to evaluate the model ability at reproducing intrusions. Potentially

this could contribute to implementing process-based constraints in climate models (Palmer et al., 2023; Regayre et al., 2023; Fasullo and Trenberth, 2012). To identify Saharan warm air intrusions, we will focus on the thermodynamic properties that define air masses formed at low-latitude subtropical desertic areas.

This study has four objectives with regard to Saharan warm air intrusions into the Western Mediterranean (WMed henceforth) during the recent historical period (1959-2022):

    (a) Generalise the definition of a Saharan warm air intrusion.

    (b) Describe the characteristics of the intrusions: spatial distribution, seasonality and trends.

(c) Assess the impact and contribution of the intrusions to extreme temperatures in the Euro-Mediterranean region.

    (d) Study which large-scale phenomena may play a predominant role in the development of Saharan warm air intrusions.

The paper, after presenting the data used for the analyses in section 2, presents a series of sections that establish the criteria to identify and assess Saharan intrusions in the WMed in the historical period. The structure is as follows: section 3 describes a methodology to identify Saharan air masses; section 4 assesses the differences between intrusions and defines different intru-
sion types through a clustering algorithm; section 5 illustrates different characteristics of the intrusions and their relationship with extreme temperatures in the Euro-Mediterranean region; section 5 proposes links between the large-scale circulation and the Saharan warm air intrusions; and section 7 shows the discussion and conclusions of the study.

## 2   Data

In this study, we use both surface and pressure-level observationally-based data in the historical period at a daily resolution. We employ daily means from 1-hourly data of the ERA5 reanalysis from years 1959 to 2022 at 0.25°x0.25° resolution (Hersbach et al., 2020). The variables used to define and identify Saharan warm air intrusions are air temperature at 925 and 700 hPa, and geopotential height at 1000 and 500 hPa. Composites of the intrusion days are computed for geopotential height and wind speeds at different levels (850 hPa, 300 hPa and 500 hPa; the latter not shown). Daily maximum surface temperature (TX) is
used to identify the relationship between Saharan warm air intrusion events in WMed (10ºW, 20ºE, 35ºN, 44ºN) and extreme temperatures in the Euro-Mediterranean region (20ºW, 50ºE 30ºN, 70ºN). In the assessment of TX and its links with Saharan warm air intrusions we also use daily mean station data from the European Climate Assessment & Data (Klein Tank et al., 2002). Table S1 shows the stations considered in the current work, following the criteria that they be representative for the Euro-Mediterranean domain and have data coverage from January 1st 1959 to December 31st 2022 (see Table S1).


## 3 Saharan air mass definition and intrusion catalogue

First we define some indicators to detect Saharan air masses. We adapt the definition of Sousa et al. (2019), which takes into account both the temperature and vertical homogeneity of a column of warm air characteristic of the latitude and surface radiative balance in the Sahara region (Webster, 2020), defined as the box 9ºW, 29ºE, 18ºN, 30ºN. Two indicators can inform us about these air mass characteristics: the geopotential thickness between levels 1000 and 500 hPa ($\Delta GH_{500-1000}$, henceforth) defined in Equation 1 and the average potential temperature between levels 925 and 700 hPa ($\overline{\theta}_{700-925}$) defined in Equation 2. Note that the latter is conserved in adiabatic heat exchanges and therefore is a good complement to the former. From Sousa et al. (2019); Galvin (2016); Webster (2020), the summer (June–August) thresholds are suggested to be around 5300 m and 40ºC, respectively, which are based on the climatological mean summer values of air masses of Saharan origin.

$$\overline{\theta}_{700-925} = \frac{\theta_{925} + \theta_{700}}{2} \; ; \text{ where } \theta = T(\frac{1000}{P})^{\frac{R_d}{c_p}} \tag{1}$$

and

$$\Delta GH_{500-1000} = GH_{500} - GH_{1000} \tag{2}$$

where $\theta_i$ is the potential temperature at the pressure level $i$ (in hPa), $T$ is the temperature, $P$ is the pressure, $R_d$ is the gas constant of dry air, $c_p$ is the specific heat at constant pressure and $GH_i$ is the geopotential height at pressure level $i$.

To validate the pressure levels used in the two indicators, we visualize the vertical profiles of the climatological Saharan and WMed $\theta$ and $\Delta GH$ (see Figures S1 and S2 in the Supplementary material). A distinction can be made between the $\theta$ vertical profiles in the warm and cold months, as $d\theta/dz$ becomes closer to zero between levels ~925-700 hPa during the warmer season (April-October). During the cold months, although the $\theta$ magnitudes between WMed and Saharan are still distinct, the $\theta$ lapse rates are similar. Therefore we keep the levels between 925 and 700 hPa for $\overline{\theta}$ throughout the cold months but argue that the choice of levels could be more flexible. In terms of the levels defining $\Delta GH$, we part from the knowledge that the relevant tropospheric variations at daily scale happen from ~500 hPa to the surface (Holton and Hakim, 2013). Then we analyse the $\Delta GH$ climograms for the Sahara and WMed (See Figure S2) and find that $\Delta GH$ 500-1000 hPa presents a good distinction between the regions and a defined seasonal variability. Sensitivity studies to the suggested levels will be presented in following paragraphs, after the identification algorithm is explained.

By conducting a study of the monthly climatologies of the two indicators defined in Equations 1 and 2, we find that the air masses over the Sahara desert behave differently in the warm and cold months (see Figure S3). The indicators in the Sahara region during the warm months have distinct (maximum) values compared to any surrounding region (the Atlantic, the Mediterranean and the equatorial band show lower values). During the cold season, the climatological $\Delta GH_{500-1000}$ and $\overline{\theta}_{700-925}$, are not as distinct in the Sahara region, but rather have an almost constant latitudinal gradient in the longitudinal band 15ºN-35ºN,30ºW-50ºE (e.g. in winter, the East Atlantic at the same latitudes as the Sahara has comparable $\overline{\theta}_{700-925}$, and $\Delta GH_{500-1000}$ values). We argue that the lesser radiative forcing in the cold months and the shorter days do not allow the

desert areas to heat the troposphere above them as notably as in the warm months, however, the indicator magnitudes over the Sahara are still distinct from those in the WMed region.

We take as representative of the Sahara desert the red box in Figure 1, which leads to climatologies close to those defined by other studies (Galvin, 2016; Sousa et al., 2019; Webster, 2020). Figure S4 shows the pool of data from all points within the Saharan and WMed (defined as the blue box in Figure 1; 10ºW, 20ºE, 35ºN, 44ºN) for all days between 1959 and 2022, which indicates the monthly historical values of the indicators in each region and shows how far away they are from each other. In Figure S4 we see that the monthly-mean Saharan air mass, represented by the mean $\Delta GH_{500-1000}$ and $\overline{\theta}_{700-925}$ (red lines), is

an extreme in the WMed distribution.

As we aim to identify Saharan air masses for other seasons beyond the summer, and we want to take into account the seasonal cycle of the indicators, we find daily thresholds for $\Delta GH_{500-1000}$ and $\overline{\theta}_{700-925}$ by computing their climatology over the Saharan region for each 31-day rolling window of the year. Different lengths of rolling windows (from 15-day to 3-month) were studied (not shown) and the results are sensitive to this length. The choice of 15 days was very restrictive, while three months was too permissive. After exploring the results we chose a 31-day rolling mean, which, although arbitrary, is an educated guess

from observing the captured events.

For an example of Saharan warm air intrusion day in May of 2015, thresholds are indicated in dashed green and fuchsia lines in Figure 1. When a grid cell exceeds both threshold values, the area is painted orange to represent a Saharan air mass. The rolling thresholds vary smoothly and allow the identification of continuous events along the year. To consider any day as an

intrusion day, the Saharan air mass (orange shading in Figure 1) must cover a sufficiently large area within the WMed, which is taken to be 5% of the region. The dashed black box in Figure 1 is a representation of how much area is 5% of the WMed is, but note that it can take any shape.

The sensitivity to the levels used to define the indicator thresholds is evaluated for the lower bounds of the geopotential thickness ($\Delta GH_{500-1000}$ and $\Delta GH_{500-850}$) and mean potential temperature ($\overline{\theta}_{700-925}$ and $\overline{\theta}_{700-850}$).In general, using a higher

lower-level potential temperature bound leads to more and lengthier events, and a higher lower-level geopotential bound leads to a very slight decrease (increase in JJA) in the amount and length of the events. The increase in events due to taking 850 hPa potential temperature might be related to the diminished diabatic cooling of the air mass as the lower bound of the indicator is further away from the surface. Changes due to $GH$ lower bound are less obvious and we can't provide a robust hypothesis for their changes. Although some changes appear in the amount of days recorded as intrusions (not shown), the captured intrusions

remain similar and long events are always captured with some slight changes in the duration. If we study the impacts in extreme temperatures and the atmospheric circulation associated with intrusion events, the results and conclusions remain the same as the ones conveyed in the subsequent sections of this work.

During the aforementioned sensitivity studies (of the levels used for the indicators and the rolling window climatologies)

we performed a visual inspection of all the event days to help in the definition of the algorithm parameters and make sure that the detected events are not spurious air masses formed away from the Sahara. All Saharan warm air intrusion events recorded with the algorithm represent continuous air masses that are displaced northward. It is noteworthy that, in line with the results

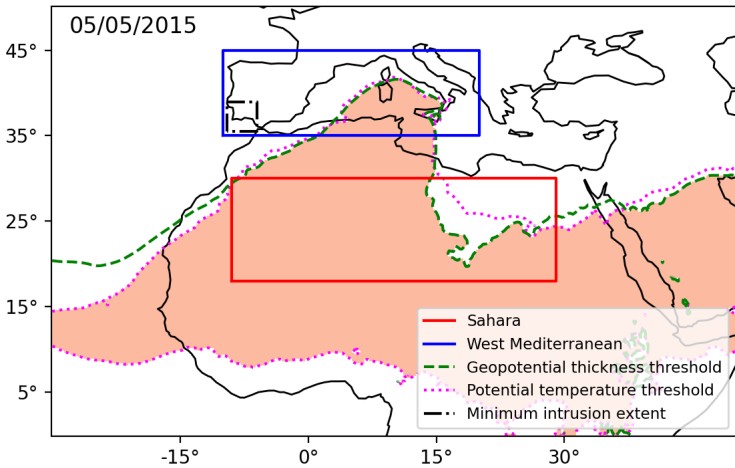

**Figure 1.** The map shows the intrusion situation of May 5th 2015, when a Saharan air mass entered along the western boundary of the WMed. The WMed (blue, 10ºW, 20ºE, 35ºN, 44ºN) and the Saharan regions used to compute the thresholds (red, 9ºW, 29ºE, 18ºN, 30ºN) are enclosed by boxes. The green (dashed) and fuchsia (dotted) lines are the boundaries of the region where the geopotential thickness between 1000-500 hPa and average potential temperature between 925-700 hPa, respectively satisfy the criteria for the air mass to be considered a Saharan air mass. The shaded orange region is the area where the two thresholds are met simultaneously. The dot-dashed black box illustrates the 5% of the WMed area, which is the minimum area inside the WMed that must be under Saharan air conditions to consider a day as an intrusion day. This 5% area can be located anywhere and in any shape within the WMed, the box is just to give an idea of its magnitude in the context of the relevant regions.

obtained for the differing Saharan characteristics in the climatologies of the warm and cold months, intrusions are qualitatively different in May-October than in November-March. In general, during the warm months, narrow bands of Saharan air are
advected northward while in the cold months intrusions tend to span a broader longitudinal band that moves in a wavelike pattern with the crest of Saharan warm air reaching the WMed. Note that, as mentioned in previous paragraphs, in the cold months, warm air advections might have mixed origins between the East Atlantic (below ∼30 ºN) and the Sahara, but always with Saharan-like warm air properties.

Through this identification algorithm, a catalogue (Figure S5) is obtained for the 1959-2022 period. A synthesis of the catalogue
is displayed in Figure 2, where the yearly amount of intrusion days per month and season can be seen (panels e-h), together with the full period's seasonally recorded intrusion days and mean event duration (panels a-d). Results show that the months with most of the intrusions are July and August, as well as December. There is also some activity during January-March and May-June. During early spring and autumn the amount of intrusions is reduced. In DJF, JJA and SON, there is a statistically significantly positive trend in the number of intrusion days per year and season (blue bars in Figure 2) with a 95% confidence
level, although it is only statistically significant for JJA when considering a 97.5% confidence level. The orange bars in Figure 2 show the mean persistence of the events, which is quite constant for all seasons.

We now focus on a particular event from this catalogue for a better understanding of the vertical properties of the Saharan

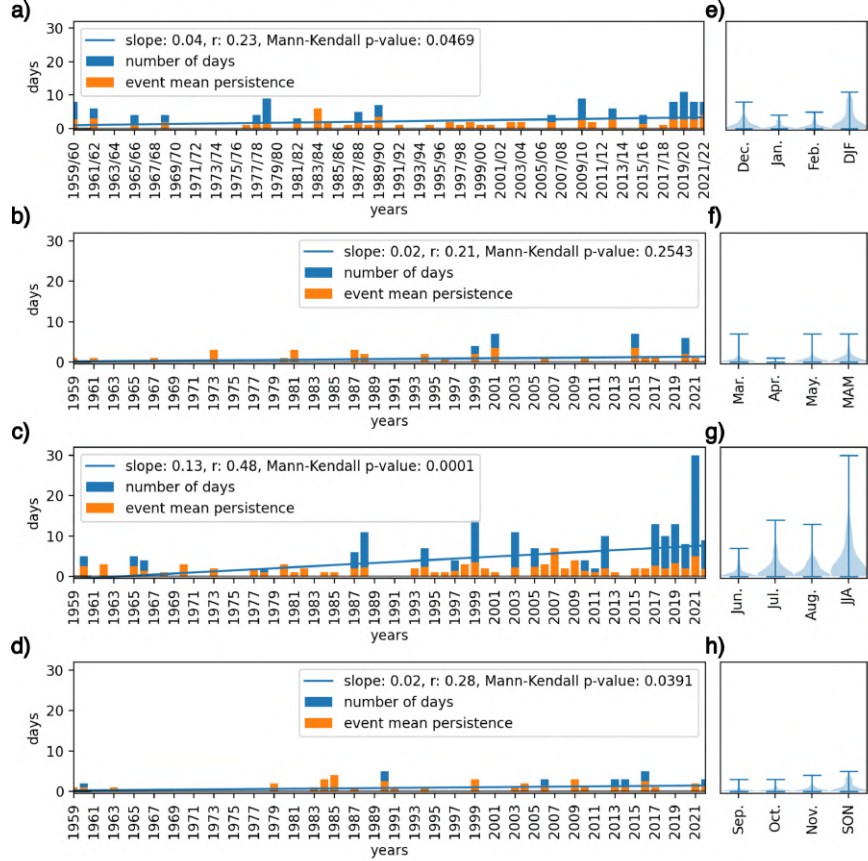

**Figure 2.** Number of identified Saharan warm air intrusions in a) DJF, b) MAM, c) JJA and d) SON per year between 1959-2022 (blue bars). The mean duration of each intrusion event, from the first to the last continuous day, is shown with orange bars. Note that when a single event is recorded for a specific year the blue and orange bars are the same size, therefore, only the orange is shown. A least squares linear regression fit to the seasonal number of intrusion days is shown in blue. e-h) show the violin plots containing the annual intrusion days for each month and season.

air masses that move into the WMed. Figure 3 shows the daily mean fields and vertical sections during an event in August of 2006. From Figure 3, we can see how a mass penetrates south of Sicily (37.5º N 17º E) driven by an intense trough in the North Atlantic. The southern and southwestern winds the 17th of August in the north of Tunisia result in a north/northeastward advection of the Saharan air mass. The vertical profiles of potential temperature show that the intruded mass has distinct features with respect to the surrounding areas. In the regions where Saharan air masses are identified, the vertical profiles show high potential temperatures, indicating that the mass comes from lower latitudes (or a region where it could warm up); the vertically-homogeneous daily-mean potential temperature in the latitudinal and longitudinal cross-section indicates the predominance of mixing by convection and diabatic heating. This agrees with the properties of air masses formed above desertic areas, where the surface long wave radiation can lead to convection, sensible heating and to a very warm and homogeneous air

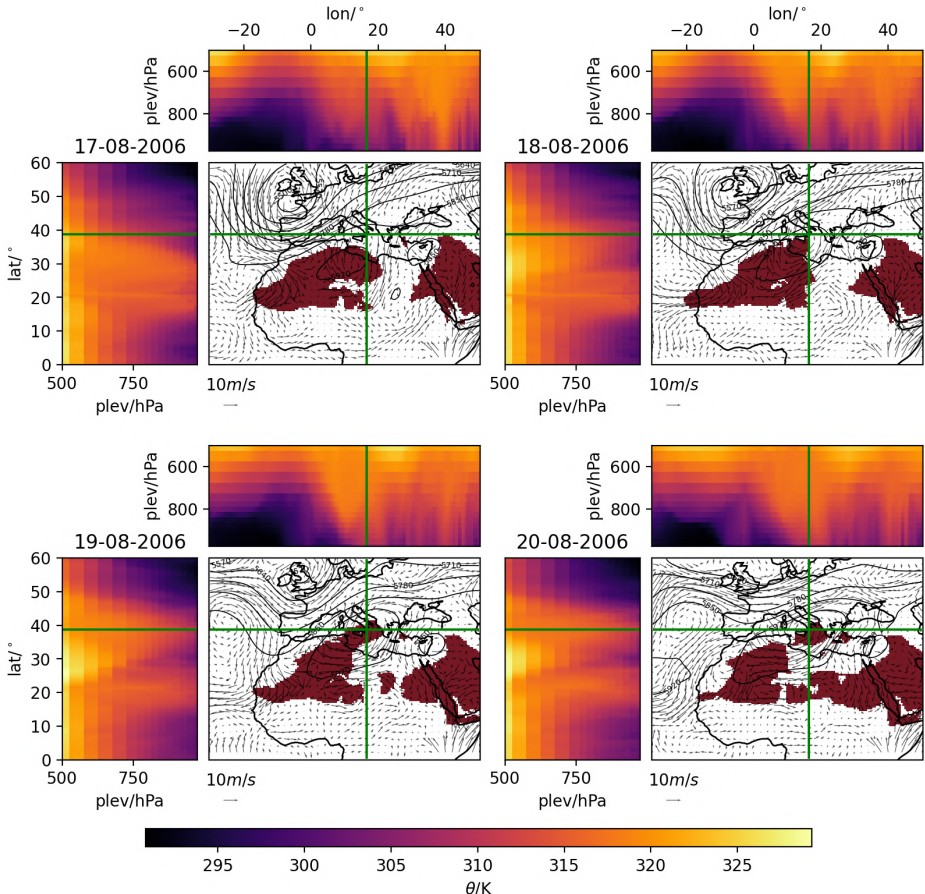

**Figure 3.** Intrusion event observed during the days 18/08/2006 and 19/08/2006. The days before (17/08) and after (20/08) the event are also displayed to frame the intrusion in the synoptic configuration that was occurring at the time. The air mass identified as Saharan is shaded in dark red while the contours show the geopotential height at 500 hPa and the quivers show the direction and magnitude of the wind at 850 hPa. The sub-panels to the left and top of the maps correspond to the vertical sections between pressure levels (plev) 975-500 hPa of potential temperature along the vertical and horizontal green lines.

column (Webster, 2020). Note that the warm air mass seems to be tilted in the vertical but that it is not considered as a Saharan air mass and it does not have direct effects on the surface temperatures. We acknowledge that such a warm upper-troposphere air mass could have indirect effects in surface temperatures due to cloud formation and radiative effects, but analysing this falls out of the scope of the current work. The reader is referred to Supplementary Figures S6 and S7 that show two more events that also illustrate the properties of a Saharan air mass.

## 4 Intrusion types

The visual inspection of individual events is informative, but lacks some generalisation to allow a climatological study of the events. Therefore, we aim to classify the Saharan warm air intrusions according to their different properties. The intrusions present local features that span smaller spatial scales than the whole WMed. This would suggest that different types of intrusion might exist depending on the area where they develop in the study region.

A clustering methodology is employed to make an initial classification of the different intrusions. To account for differences in the radiative forcing, atmospheric circulation and thermodynamics in the Sahara along the year (as seen in Section 3), we apply a clustering analysis of the intrusion events in each meteorological season (DJF, MAM, JJA and SON, henceforth). Our approach inputs the $\Delta GH_{500-1000}$ and $\overline{\theta}_{700-925}$ variables during intrusion days in the WMed region into a clustering procedure. The method is a combination of a reduction of the dimensionality via S-Mode principal components analysis (99% of the spatial variance is maintained) and the k-means clustering of the matrix of stacked indicators (Tencer et al., 2016; Muñoz et al., 2015, 2016, 2017; Olmo et al., 2024).

A sensitivity analysis is performed to determine the optimal number of clusters, using both the Silhouette score (Rousseeuw, 1987) and the Pseudo-F metric (Calinski and Harabasz, 1974). The Silhouette score is calculated for each data sample based on the mean intra-cluster distance (the average distance within a cluster) and the mean nearest-cluster distance (the average distance to the nearest cluster). The Pseudo-F score is computed by dividing the between-cluster sum of squares (which measures the dispersion between cluster centroids) by the within-cluster sum of squares (which measures the dispersion of data points within their assigned clusters). Together, these metrics provide insight into the clustering structure. However, neither metric indicates a clear optimal number of clusters, likely due to the high variability of the indicators during intrusion days. To address this issue, we prioritise relative maxima in the Silhouette scores that coincide with an elbow in the Pseudo-F curve, which allows us to synthesise as much as possible the number of clusters (intrusion types, IT). The resulting number of ITs are 5 for DJF, 3 for JJA and 4 for MAM and SON (see Figure S8). Each intrusion day is assigned to a centroid according to the Euclidean distance between the indicator fields of each day and the centroid.

Results obtained from clustering the intrusion days for $\overline{\theta}_{700-925}$ and $\Delta GH_{500-1000}$ over the WMed show that the ITs can be distinguished by the location where the anomalously warm air mass is found and in its intensity (how big the anomaly magnitudes are). Figure 4 shows the $\Delta GH_{500-1000}$ and $\overline{\theta}_{700-925}$ composites for the JJA ITs. They show a distribution between western, central and eastern WMed, and IT2 represents more intense intrusions (larger $\overline{\theta}_{700-925}$ and $\Delta GH_{500-1000}$ positive anomalies). Figure S9 shows the rest of the seasons. MAM and SON display similar characteristics but dividing the region into four longitudinal bands; intensities and extent of the positive anomalies vary across ITs as well. DJF also has longitudinal separations, and, apart from displaying different intensities and extent, some latitudinal differences appear, suggesting that different ITs are also clustered in terms of how much they can move northward. We introduce the term "area of influence" of each IT as the green contours seen in Figures 4 and S9, which are defined as the limit where intrusion events have been recorded for a specific season and IT.

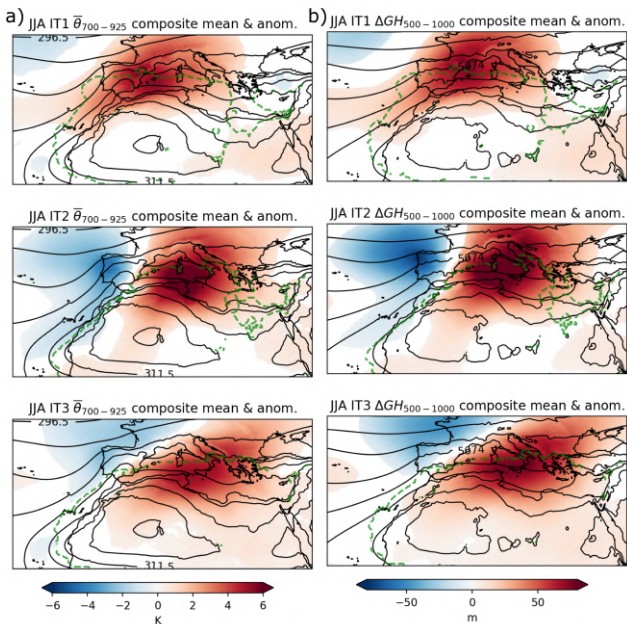

**Figure 4.** Composites of the three June-to-August ITs of (a) $\bar{\theta}_{700-925}$ and (b) $\Delta GH_{500-1000}$ anomalies with respect to the mean climatology of the days without intrusion. No values are painted (white) where the anomaly is not significantly different from 0 with a t-Student test and 95% confidence level. The solid (black) contours display the climatological values of $\bar{\theta}_{700-925}$ and $\Delta GH_{500-1000}$ from JJA days without intrusion. The area of influence is displayed with a green (dotted) contour, and represents the limit where Saharan air masses have been recorded in the historical period for a specific IT.

Figure 5 shows the JJA intrusion days colour-coded according to the IT they belong to (see Figure S10 for the other seasons). During persistent events, the IT can either remain constant or transition, meaning that the Saharan air mass can remain stationary in the IT area of influence or move towards other regions.

Given that intrusion days can shift from an IT to another, the transition probability heatmaps are computed (see Figure S11). They inform about the persistence or changes in the IT between one day and the next one. ITs are generally persistent in JJA, although IT1 and IT2 can often transition to IT3. DJF shows very little persistence, and MAM and SON show mixed behaviours depending on the IT.

We want to take advantage of the separation of intrusion days into different ITs to see if there is any connection between Saha-
225 ran warm air intrusions and any of the relevant teleconnection indices in the Mediterranean region. There are many large scale drivers that affect the climate of the WMed, some that might be interesting to assess their link with the ITs are: i) The North Atlantic Oscillation (NAO) and it's summer expression (SNAO), which are the leading modes of large-scale atmospheric variability in the North Atlantic and are defined by the pressure gradient between Iceland and Azores (Hurrell, 1995), and through Principal Component Analysis of the summer mean sea-level pressure (Folland et al., 2009), respectively. It has a direct effect
on the position and strength of the subpolar jet and therefore affects the circulation, weather patterns and different atmospheric

variables in the Euro-Mediterranean (Hurrell, 1995; Folland et al., 2009; Bladé et al., 2012a). ii) The Western Mediterranean Oscillation (WeMO) is designed to consider the atmospheric dynamics of the Western Mediterranean by taking the pressure gradient between the gulf of Cádiz and the Po plain (Martin-Vide and Lopez-Bustins, 2006). iii) The Arctic Oscillation (AO) (Thompson and Wallace, 1998) is the hemispheric leading mode of variability from surface to stratospheric levels (Dunkerton and Baldwin, 1999; Gerber et al., 2010) and it mainly drives the fluctuations of the subpolar jet. iv) Atlantic Multi-decadal Variability (AMV, Deser et al., 2010) is the low frequency fluctuation of the North Atlantic sea observed in surface and subsurface variables. We use the definition of the Atlantic Multi-decadal Oscillation (AMO) to capture this low frequency variability mode (Kerr, 2000; Klotzbach and Gray, 2008), which intends to distinguish itself from the NAO-influenced tripole pattern at interannual time scales (Enfield et al., 2001). It is defined as the detrended average anomalies of sea surface temperatures in the North Atlantic basin (detrending is performed by subtracting the mean global SSTs). The AMO has been linked to Mediterranean temperature fluctuations (Mariotti and Dell'Aquila, 2012).

Kendall-tau ($\tau$) correlations have been computed between the yearly intrusion days of every IT (labelled as ID) and the aforementioned teleconnection indices. While the association values are low, statistically significant relationships at the 95% level have been found for the Atlantic Multidecadal Oscillation in summer, the Arctic Oscillation in winter and the Western Mediterranean Oscillation also in DJF. The results suggest that a warmer-than-usual North Atlantic could enhance the occurrence of Saharan warm air intrusions in the central and western part of the WMed during JJA; In DJF, having a negative AO (predominance of a low pressure in the northwestern WMed) is correlated with an enhanced number of intrusion events; having a relatively high pressure in the south western part of the WMed with respect to the northeastern part of the WMed (positive WeMO) seems to also favour Saharan warm air intrusions in DJF. The Kendall-tau correlation magnitudes are:

$$\tau(ID_{\text{IT2}}^{\text{JJA}}, AMO_{\text{JJA}}) = 0.26$$

$$\tau(ID_{\text{IT3}}^{\text{JJA}}, AMO_{\text{JJA}}) = 0.20$$

$$\tau(ID_{\text{all IT}}^{\text{DJF}}, AO_{\text{DJF}}) = -0.20$$

$$\tau(ID_{\text{all IT}}^{\text{DJF}}, WeMO_{\text{DJF}}) = 0.22$$

## 5 Intrusions and extreme temperatures in the Euro-Mediterranean region

In this section we focus on the broader Euro-Mediterranean region (EM, henceforth) and assess the influence of intrusions on extreme temperatures. We compute extreme temperature days for the historical period at each grid cell, estimated as those days that exceeded their daily maximum surface temperature 7-day rolling window climatological 90th percentile (TX90p, Zhang et al., 2005 Wang et al., 2013). The same is done for station data from the ECA&D daily mean temperature dataset (see list of stations in Table S1). We aim to quantify the probability of getting an extreme temperature day in any region when a certain IT happens. We define the impact on extreme temperatures for each IT and season in Equation 3 as:

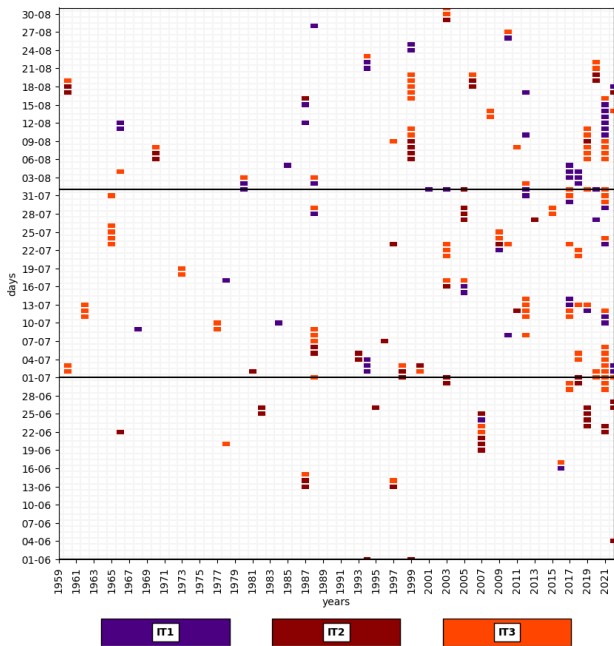

**Figure 5.** Historical catalogue of intrusion days in the period 1959-2022 (x-axis) and from the 1st of June to the 30th of August (JJA, y-axis). The days identified as intrusions are coloured with their assigned IT. Black lines indicate the change of months.

$$\text{Impact} = P(TX90p \mid \text{IT}n \text{ intrusion day}) = \frac{P(TX90p \cap \text{IT}n \text{ intrusion day})}{P(\text{IT}n \text{ intrusion day})} \tag{3}$$

where $P(TX90p \mid \text{IT}n \text{ intrusion day})$ is the probability of having an extreme temperature day (TX90p) given that the day is

also an intrusion day (i.e. the intrusion impact) from a specific IT (the $n$ in IT$n$ labels all possible ITs in each season).

The $P(TX90p \mid \text{IT}n \text{ intrusion day})$ of Saharan intrusions on temperature extremes in the EM has different spatial extents and probability magnitudes, and depends on the season and IT (see Figure 6). The results highlight the Saharan warm air intrusions impact, which can reach probabilities from 60 to 100% in the area of influence. During DJF, MAM and SON, Saharan intrusions can have a notable impact on regions beyond the WMed, meaning that intrusions have an effect outside the area of

influence where intrusion days were recorded. Contrarily, during JJA, the impact is quite confined to the area of influence of each IT. To see if this can be due to the troposphere dynamics being slower in summer, we assess the results for the first and second days after the end of the intrusion events to check if the impact is switched on after the events end (see Figure S12 for the first day). There is some impact on extreme temperatures although lower than during intrusion days. Also, the impacts after the events end are generally shifted eastward from the area of influence of each IT, in line with the climatological westerly

circulation in the region. Therefore, Figure S12 disproves that JJA intrusions might move slower and have an impact outside the area of influence the days after an intrusion event. This result suggests that if JJA has no impact outside the area of influence it is because Saharan warm air intrusions, and their impacts on extreme temperatures, are simply more confined to the area of

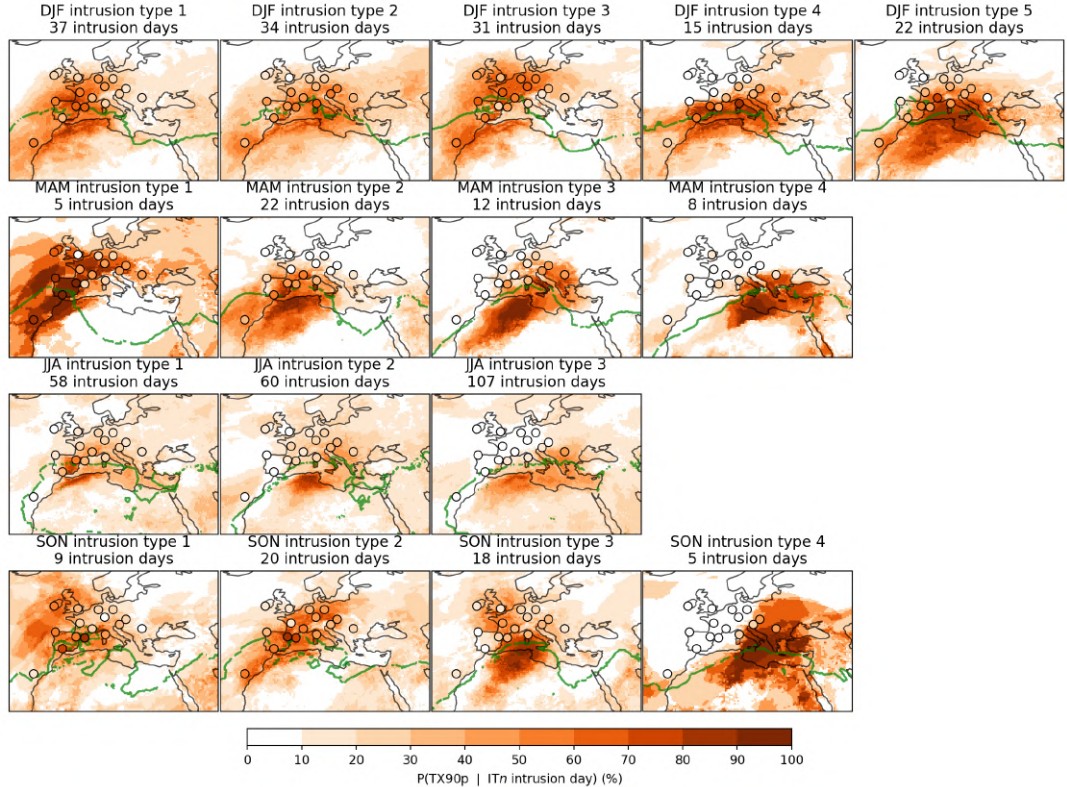

**Figure 6.** Impact of the Saharan warm air intrusions on extreme temperatures in the EM region for the different seasons (rows) and ITs (columns). The impact is measured as the percentage of intrusion days that coincide with an extreme temperature day (TX90p). The amount of intrusion days for each season and IT is specified in the title of each panel. Colored dots are results from ECA&D station data (Table S1). The area of influence is displayed with a green contour, and represents the limit where Saharan air masses have been recorded in the historical period for a specific IT.

influence.

The station data is generally in agreement with ERA5 results, which gives us confidence in the reanalysis. Note that some of the results come from a small number of intrusions (each season and IT number of intrusion days are provided in the top of each panel) and therefore probability results must be treated with care.

To quantify how the presence of an intrusion increases or decreases the risk of having TX90p days we compute the risk ratio (Equation 4) between the probability of having TX90p conditioned on having an IT$n$ intrusion day versus that of having TX90p conditioned on being on any other day than the IT$n$ intrusion day.

$$\text{risk ratio} = \frac{P(TX90p \mid \text{IT}n \text{ intrusion day})}{P(TX90p \mid \text{no IT}n \text{ intrusion day})} \tag{4}$$

where the numerator is the impact and the denominator is the probability of having an extreme event when the day is not an IT$n$ day. The risk ratio is the ratio of how much more risk there is to have an extreme temperature event if the day is an IT$n$ intrusion day than if it is not. Risk ratio will be closer to one when the two probabilities are similar, and therefore the effect of the intrusion is not affecting the probability of having a TX90p day. When the risk ratio has values above 1 it means that having an extreme temperature is more probable when an intrusion is present. Risk ratio under one means that having an extreme temperature day is more probable on non-intrusion days. Values above 10 suggest a strong difference in the occurrence of TX90p between having or not an IT$n$ intrusion day (Ellison, 1996).

Figure 7 displays the risk ratio and we see that results from Figure 6 are validated and that many areas in MAM and SON display very large values suggesting a strong influence on the intrusions to extreme temperature values. Some substantial values in the other seasons can be seen, and in DJF, MAM and SON there are magnitudes above one that span beyond the area of influence.

Finally, if we compute, for every season, the percentage of TX90p days that are also an IT$n$ intrusion day, i.e. P(TX90p $\cap$ IT$n$ intrusion day) / P(TX90p), we get an idea of the contribution of intrusion days to the TX90p days. We see that in some regions intrusions contribute up to $\sim$10% of the seasonal TX90p days in JJA and up to 4% in DJF (see Figure S13).

## 6  Large-scale circulation associated with the intrusion onset

Composites for each IT can be computed for any meteorological variable. This becomes helpful to understand the anomalies during intrusion days. First, we need the anomalies of any variable of interest, which are computed by subtracting the non-intrusion climatologies of 7-days rolling windows centred on each calendar day (Zhang et al., 2011; Tencer et al., 2016; Olmo et al., 2020). For example, we compute the upper-tropospheric geopotential height composites of any IT by first calculating the non-intrusion climatologies (using a 7-day rolling average), subtracting them from the intrusion days' composite and then averaging these anomalies for the intrusion days. We are particularly interested in the relationship between the circulation and the onset of the intrusion events. Therefore, the composites are from the first day of the intrusion events. The results help to identify the mechanisms behind the intrusions.

In previous analyses, we saw that the different ITs can be related, in part, to where the intrusion is located in the WMed (Figure 4). This result is reinforced by Figure 8-9,a (for DJF and JJA, respectively) and S14-S15,a (from MAM and SON, respectively), which shows the anomalous winds at 850 hPa for each IT. Masked areas mean that the anomalies are not significantly different from zero according to a two-tailed 95% confidence t-test (Ukkola et al., 2020). The northward wind anomalies coincide with the areas of influence of the ITs meaning that there is an advection of southern air into the WMed. Figure 8-9,a and S14-S15,a also show the anomalies in sea-level pressure (SLP) in shading, and for most of the ITs there is an associated low pressure located to the west, northwest or north from the area of influence. Some ITs do not have a clear statistically significant SLP anomaly signal (DJF IT3 and SON IT2). The low-tropospheric circulation and the SLP anomalies are coherent as the wind anomalies follow an (anti-) clockwise pattern around anomalous (Highs) Lows.

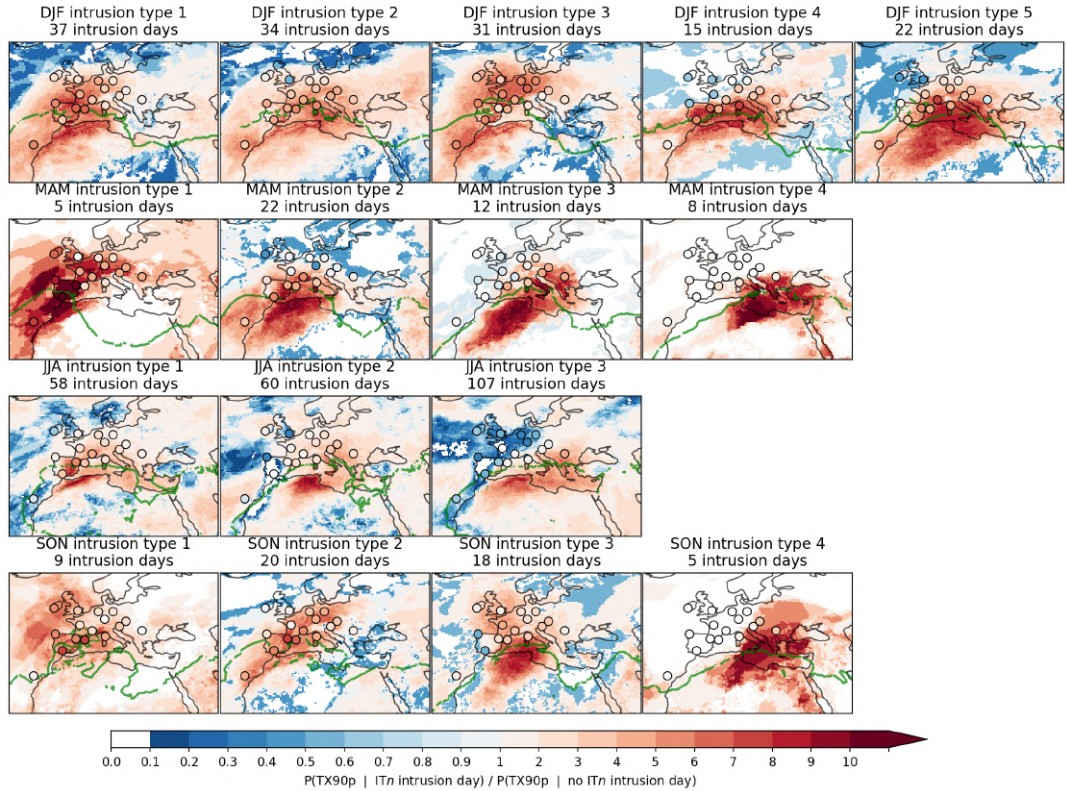

**Figure 7.** Risk ratio of having an extreme temperature day (TX90p) in days with and without Saharan warm air intrusions for each season (rows) and IT (columns). It is computed as the fraction of the impact divided by the probability of having an extreme temperature day when no ITn day is recorded. Colored dots are results from ECA&D station data (Table S1). The area of influence is displayed with a green (dotted) contour, and represents the limit where Saharan air masses have been recorded in the historical period for a specific IT.

In terms of the IT strength, we have shown how JJA IT2 is associated with larger $\overline{\theta}_{700-925}$ and $\Delta GH_{500-1000}$ anomaly composite magnitudes (Figure 4), and this is in line with larger pressure and $GH_{300}$ anomalies and a stronger northward component of the wind in the circulation composites in Figure 8. Nonetheless, this is not found in DJF, MAM and SON as there is not a relationship between strong indicator composites and the circulation anomalies composites. We therefore must remain cautious in claiming any link between the intensity of the IT and its circulation composites.

When looking into the upper-troposphere (300 hPa, but similar results are found in 500 and 200 hPa), very well defined and significant anomalies are detected both in wind speeds and geopotential height (see Figure 8-9,b and S14-S15,b). Generally, there is always a statistically significant and well defined geopotential high over the northern tip of the area of influence of the intrusion. Most times this high is accompanied by a low that is located to the west, northwest or north. This anomalous negative geopotential centre is generally well aligned vertically with the negative sea level pressure anomaly (although some ITs present some baroclinicity). The winds in the upper-troposphere are consistent with the geopotential centres, suggesting

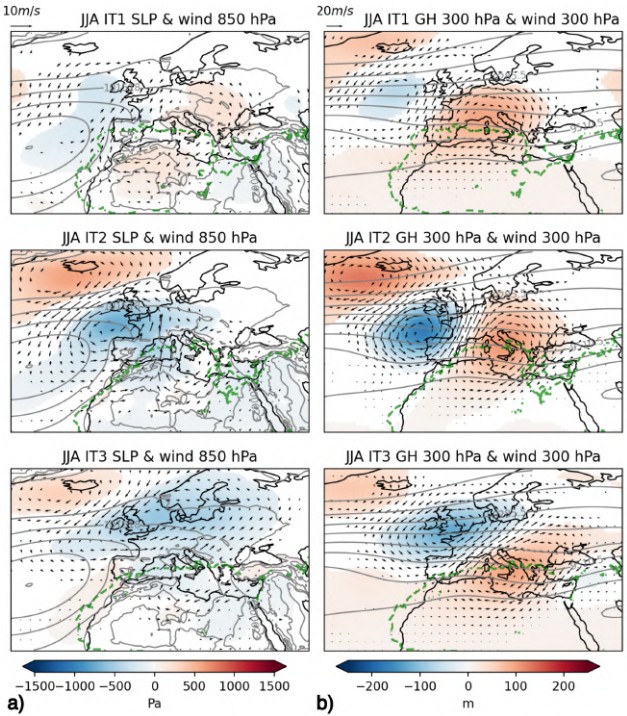

**Figure 8.** Anomaly composites of the first days of the JJA intrusion events for (a) composites of SLP (shading) and wind speed at 850 hPa (quivers) and (b) geopotential height (shading) and wind speed (quivers) at 300 hPa for the three ITs (rows). Anomalies are computed with respect to the non-intrusion climatologies of 7-day rolling windows centred on each calendar day between 1959-2022. Contours show the climatological SLP (a) and 300 hPa geopotential height (b). No values are shown where the anomaly is not significantly different to 0 with a t-Student test and 95% confidence level.

that the general circulation (westerlies) is slowed down south of the geopotential positive blob, which can lead to onsets of intrusion events. The main differences between seasons is that the anomaly centres in DJF are generally broader and more intense than in JJA. MAM and SON composites have similar behaviours and also show broad and intense anomaly centres but to a lesser extent than DJF. The composites have certain similarities to the composites of subtropical ridges proposed by Sousa et al. (2018). We argue that subtropical ridges might be a main driver of Saharan warm air intrusions, although not a necessary condition.

The composites of the days leading to the intrusion onset (Figures S16-S17) show that the anomalies in the upper-troposphere generally propagate over the North Atlantic (in DJF, with large spatial extents) and in the eastern North Atlantic (in JJA, with smaller spatial extents), first as a deepening trough (negative geopotential anomaly at 300 hPa) that leads to a high over the northern tip of the area of influence of the IT.

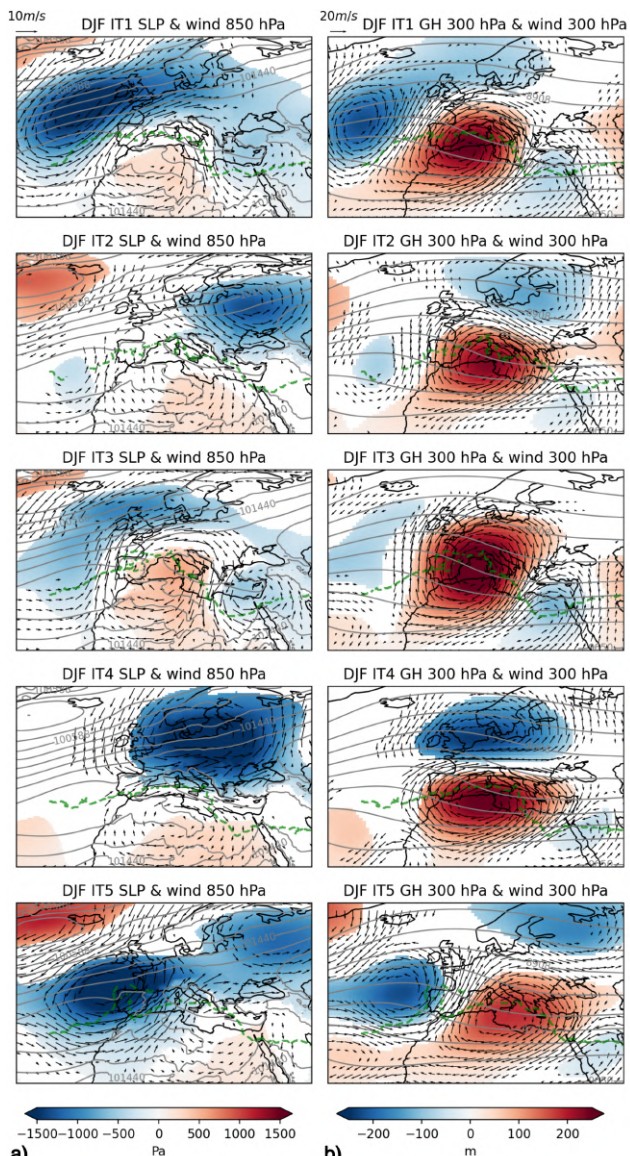

**Figure 9.** Same as Figure 8 but for the DJF ITs.

## 7 Discussion and conclusions

This study aims at characterising Saharan warm air intrusions into the western Mediterranean, as the limited literature suggests that these events significantly influence extreme temperatures in the region (Sousa et al., 2019). Our goal has been to develop a workflow to i) identify intrusion days throughout the year, ii) classify them into different types, iii) assess their impact *on* and contribution *to* temperature extremes, and iv) evaluate the large-scale conditions in the onset of Saharan warm air intrusions.

The method used identifies intrusions of warm air originating in the Sahara desert that propagate northward into the WMed. This is achieved through two indicators: the geopotential thickness between 1000 and 500 hPa, and the mean potential temperature between 975 and 700 hPa, following the work by Sousa et al. (2019). These indicators provide insights into the temperature and vertical mixing within the air column—key characteristics of air masses formed in desert regions. The interaction between the land surface and the atmosphere heats the air column, making it highly convective but without cloud formation (Galvin, 2016; Webster, 2020).

During the initial stages of our study, we observed that the tropospheric air properties over the Sahara differ significantly from those of surrounding regions between May-October (warm months) but not from November-April (cold months). The climatology of the Saharan air masses provides a threshold for identifying warm, vertically homogeneous air masses that reach the WMed, as illustrated in Figure S3. Typical Saharan air masses are found to lead to a higher probability of having extreme temperatures in the WMed (as seen in Figure 7).

A visual inspection of the synoptic configurations during all the events helped tuning the algorithm parameters and revealed that our method successfully captures continuous air masses advected northward into the WMed, increasing our confidence in the method. We argue, nonetheless, that the visual inspection does not ensure that we are missing less intense Saharan intrusions due to too restrictive $\bar{\theta}_{700-925}$, and $\Delta GH_{500-1000}$ thresholds. The atmospheric circulation leading to these intrusions is diverse, often resulting in intrusions confined to spatial scales smaller than the entire WMed. During the cold months, the climatological values of the indicators ($\bar{\theta}_{700-925}$, and $\Delta GH_{500-1000}$) over the Sahara do not seem exclusive to the Sahara, but rather have similar properties in regions outside the Sahara that are at the same latitude ($\sim$ 15ºN-35ºN,30ºW-50ºE). This is due to the reduced influence of the surface radiative balance over desert land that produces the distinctive air mass properties during the warm months. Therefore, we conclude that intrusions during the cold months could also be mixed with eastern Atlantic air masses at Saharan latitudes. This does not change the fact that Saharan-like air masses are penetrating the WMed, but it is important to note that the provenance can be non-exclusively from the Sahara. Also, from the qualitative assessment of some cold month intrusions (see Figure S6 for an example), we see that cold month intrusions can behave like a broad subtropical ridge with a larger longitudinal extent than warm month episodes. This result is backed by the circulation composites (which will be further discussed in next paragraphs) in DJF, where the upper-troposphere positive anomalies are generally broader than in JJA (see Figures 8 and 9); in agreement with the results in Sousa et al. (2018).

This study provides, for the first time, a comprehensive catalogue of Saharan warm air intrusions throughout the year, covering the period from 1959 to 2022. Our approach, while simple and based on just two variables, effectively identifies intrusion events in the WMed. Other studies, some not necessarily focused on the Sahara, have used tracking algorithms that may offer greater precision in detecting Saharan air intrusions, though at a significantly higher computational and data volume cost (Lemus-Canovas et al., 2021; Fix et al., 2024; Wernli and Davies, 1997; Santos et al., 2015; Bieli et al., 2015; Schielicke and Pfahl, 2022). We argue that our computationally cheap workflow can easily be ported to analyse climate model output, which is a key step to evaluate their performance in reproducing this mechanism.

Given that the spatial scale of the intrusions is smaller than the WMed and the visual inspection of the events hinted that intrusions have diverse behaviours, we applied a clustering method to identify distinct intrusion types (ITs). This approach yielded

five types for DJF, three for JJA and four for both MAM and SON, each representing the area of influence where the intrusions occur, effectively dividing the study region into subregions, and also discriminating the intensity and extent of the Saharan warm air intrusions. These distinctions are relevant because different ITs have varying impacts on regional temperatures and are associated with different large-scale circulations, as discussed below.

While ITs can evolve during a single event, the most common behaviour is for an IT to either maintain its type or transition to a relatively eastern IT, consistent with the prevailing westerly circulation in the region (Olmo et al., 2024). We also observed a statistically significant increase in intrusion days over the historical period for JJA, DJF and SON, compatible with long-term thermodynamic changes and the warming and expansion of the troposphere that can make our climatological definition of Saharan air masses reach higher latitudes more easily or even be generated more to the north. These thermodynamic changes have been extensively linked to global warming (Staten et al., 2018; Webster, 2020; IPCC, 2023), although it is important to consider how climate variability at different timescales can affect temperature increases in the Sahara and WMed regions (IPCC, 2023). Global warming and climate variability could also modify the circulation dynamics (Coumou and Rahmstorf, 2012; Rousi et al., 2022) and directly influence the frequency of mechanisms that lead to Saharan warm air intrusions. This attribution falls out of the scope of the current work but we believe such a study should be performed in the future to improve intrusions' predictability. The trend in intrusion days is also reflected in regional observations of extreme heat events in Iberia (Del Río et al., 2011; Cardoso et al., 2019; Fonseca et al., 2016; Sandonis et al., 2021), the Italian peninsula and surrounding islands (Scorzini and Leopardi, 2019; Monforte and Ragusa, 2022; Bey et al., 2024; Caloiero and Guagliardi, 2020), and the broader Mediterranean region (Campos et al., 2024; Cos et al., 2022). Of particular relevance is the summer of the year 2021, when the amount of intrusion days reached 30 days (more than five standard deviations higher than the mean in summer). The summer of 2021 has been studied for its extreme temperatures in the Euro-Mediterranean region (Demirtaş, 2023; Founda et al., 2022).

The contribution of Saharan warm air intrusions in the WMed during the historical period is important. We highlight a key metric of the relationship between extreme temperatures and intrusions: the impact. We define it as the probability of having an extreme temperature day (TX90p) given that the day is also an intrusion day. The results show that intrusions are very impactful in terms of temperature extremes within their specific IT's area of influence, and further north and northwest into the EM region (with exception of JJA). The risk ratio of having extreme temperatures if there is an IT$n$ day or not has also been computed and we see that there is a stronger risk of having extreme temperatures in broad areas of the EM region for all seasons when intrusion days are recorded. In terms of contribution, i.e. how many TX90p days coincide with an intrusion day, JJA stands out as the only season with a notable contribution of around 10% of extreme temperature days, primarily due to the higher frequency of intrusion events during this period. These findings confirm that Saharan warm air intrusions have an important impact on extreme temperatures across all seasons. However, because the frequency of intrusions varies seasonally, only those in JJA make a relevant contribution to overall extreme temperatures. This initial study has not considered the radiative impact of Saharan aerosols to extreme temperatures, as in Sicard et al. (2022); Cuevas-Agulló et al. (2023); nor has assessed the evolution of Saharan air masses during intrusion events, similar to what is done in the analysis of backward trajectories of extreme temperature events as in Sousa et al. (2018); Zschenderlein et al. (2019); Mayer and Wirth (2025).

Through an analysis of anomaly composites for each intrusion type, we identified distinct upper-tropospheric circulation anomalies. The composites of upper-tropospheric geopotential show an anomalous high over the area of influence of the ITs, associated with an anomalous circulation around that high that slows down the main flow over Northern Africa. Furthermore, we see that the anomalous high is normally preceded by a westerly shift of positive and/or negative geopotential height centres in the North Atlantic. This pattern holds for all seasons. In some cases, the days leading to the onset of the Saharan warm air intrusion hint at the propagation of an anomaly wavetrain across the North Atlantic. This suggests that there might be an upstream source that forces the anomalous high over the WMed. Some work in this direction but related to other phenomena has been recently conducted by Sandler et al. (2024). They suggest that the connection between upper-level geopotential anomalies and surface processes in the Mediterranean can originate from Atlantic Ocean sea-surface anomalies. Teng et al. (2022) identify a positive trend in the upper-tropospheric geopotential height in JJA over the EM, associated with an hemispheric wave-like structure forced by the North Atlantic. Therefore, trends in JJA Saharan intrusions could be influenced by the circulation trend described in Teng et al. (2022). This offers an explanation that links the upper-troposphere circulation and the Saharan intrusions. Other research has examined the role of the polar jet on extreme temperatures (Rousi et al., 2022; Wang et al., 2013), though the mechanisms behind its fluctuations remain unsolved. It would be relevant to understand, how the synoptic configurations specific to each intrusion event are related to fluctuations in the westerly eddy-driven jet stream over the North Atlantic, including anticyclonic/cyclonic wave-breaking processes in line with the work of Sousa et al. (2021); Woollings et al. (2010). Such an analysis of the large-scale dynamics falls beyond the scope of the present work. Further investigation is needed to identify the forcings driving the large-scale circulation anomalies and explore the potential role of teleconnections. Gaining a clearer understanding of these processes could improve the predictability of Saharan warm air intrusions and enhance the evaluation of model simulations, ensuring confidence in their representation of this important phenomenon and its impacts.

*Code and data availability.* The data used can be accessed through the Climate Data Store maintained by Copernicus at: https://cds.climate.copernicus.eu/datasets.
The code developed to identify the Saharan warm intrusions and to compute all other results of this study can be found at: https://doi.org/10.5281/zenodo.14925526

*Author contributions.* PC, RM and FD designed the study. PC developed the diagnostics, and wrote the initial manuscript. PC, MO, DC, LP and AM designed the clustering algorithm. All authors contributed to the interpretation and discussion of the results and the improvement of the manuscript.

*Competing interests.* The contact author has declared that neither they nor their co-authors have any competing interests.

*Acknowledgements.* We wish to thank all those who have provided the data used for this work and for the data support by M. Samso. We would also like to thank the comments and suggestions from the two anonymous reviewers who greatly contributed to the improvement of this study. PC would like to especially thank J. Mindlin for her invaluable presence, random brilliant thoughts and discussions throughout this study. This work was partly supported by the National Research Agency (AEI-Agencia Nacional de Investigación) through the project GLORIA (TED2021-129543B-I00). MO is funded by the AI4Science PN070500 fellowship within the "Generación D" initiative, Red.es, Ministerio para la Transformación Digital y de la Función Pública, for talent attraction (C005/24-ED CV1). Funded by the European Union NextGenerationEU funds, through PRTR.

450

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
