# Peer review of "Saharan warm air intrusions in the Western Mediterranean: identification, impacts on temperature extremes and large-scale mechanisms"

_EGUsphere, 2024_

## Referee Comment (RC1)

**Review of: "Saharan warm air intrusions in the Western Mediterranean: identification, impacts on temperature extremes and large-scale mechanisms" by Cos et al.**

**Summary and General Assessment**

Warm air from the Sahara can be advected to the Mediterranean and possibly cause extreme temperatures in the Mediterranean and beyond. Cos et al. use an identification method based on the geopotential thickness between 1000 and 500 hPa, and the mean potential temperature between 925 and 750 hPa to investigate the so-called Saharan warm air intrusions into the Western Mediterranean. Using this method, they calculate a catalogue of past intrusions. The authors perform a clustering of intrusion days based on the distribution of geopotential thickness and mean potential temperature, and find three to five distinct clusters per season, that mainly differ in their geographical location. The transition probabilities between the clusters, the correlations with the teleconnection indices, and the composites of meteorological fields are computed per cluster per season. Extreme temperatures are often observed during intrusion days. Based on the conditional probabilities the authors argue that intrusions have a great impact on heat events, but not such a big contribution, i.e. intrusions often cause heat, but heat is not always caused by intrusions. The composites of the meteorological fields show that the different locations of the clusters are due to different locations of the driving pressure patterns.

The study is motivated by increasing temperatures in the western Mediterranean and the mention of such intrusions in the literature. While the connection between the intrusions and heat is not too surprising, the concept of Saharan warm-air intrusions into the western Mediterranean gives a new perspective on extreme temperature events in the region.

The overall structure of the study is well-organised, however some parts need improvement as they are not so easy to follow. The figures support the presented results mostly well, but their readability could be improved. Below I have major, minor, and figure-specific concerns, comments, and questions that should be addressed.

**Major comments**

*Section 3*
In this section, you describe the algorithm used to identify the intrusions, based on the two indicator variables $\Delta GH_{500-1000}$ and $\overline{\theta}_{750-925}$. You refer to the work of Sousa et al. (2019), however, I still lack a motivation/explanation for why these levels are useful to determine the air mass. Especially considering that daily and seasonal variations may change the height of the boundary layer height drastically, for example. Have you looked at how sensitive the results are to your choice of indicators?

*l 91-105*
This paragraph is very difficult to follow. It can be improved by rewriting/rephrasing, but since it is a crucial paragraph to the paper, I consider this a major comment. Let me summarise what I understood:
Monthly climatologies of $\Delta GH$ and $\overline{\theta}$ show that in summer values are maximised over the Sahara, but during the rest of the year they decrease slowly and constantly between 15 and 35°N.
Then, you choose a region as representative for the Sahara. You average over this region, which was chosen so that you meet the values described in literature. I am missing the exact extent of this region in the text.
Then, you calculate 31-day rolling means, averaged over the region, to define thresholds. Why do you chose 31d in particular and how sensitive is your result to this choice?
The air mass is then defined as all grid cells that exceed both threshold values. Does this always yield a coherent air mass as depicted in Fig 1? I would imagine there might be single cells within the air mass that are not identified, or single cells outside of it that are?
Please improve this paragraph by explaining step by step what you did and why.
In lines 109-115 you mention again that in the colder seasons, the values in the Sahara are not as distinct, since there is a constant gradient present. First, this is a little bit repetitive to what was mentioned before (l 92 following). Second, you do not really explain what the consequences of this are. I would imagine that the sensitivity to the threshold values varies a lot across seasons because of this? Does this mean your method works better in summer than in the rest of the year?

*l 107*
"air masses that are displaced from the Sahara". Does that mean the air mass has to cover parts of the Sahara at all times to ensure the origin?
Is the visual inspection necessary for your detection algorithm in general, or was this just a proof of concept in this case? Because if there is doubt about the origin of the air masses, and this step is always necessary, this is a big issue in the applicability of the method to other cases.

*l 116-118*
As far as I understand, the development of the catalogue and analysis of seasonal and inter-annual variations is a crucial part of your study. I was therefore disappointed to be referred to the Supplement for such a central information. Why don't you show an average number of events per month and their spread across years (e.g. as a boxplot). Then your description of the seasonal variability can refer to this plot.

*Fig 3*
From the cross-sections it becomes obvious that the air mass tilts up towards the edges, especially in the North and West. This will not be captured by your methodology, which focuses on the lower levels only. I think, however, this warm, dry air mass might still have considerable effects in those regions. This goes in the same direction as the question about how sensitive your results are to the choice of the indicator variables and their respective levels. Could you discuss this a little bit further?

*l 153 and following and Fig 4*
I was wondering how similar the clusters between the seasons are. The different colour scales make it difficult to compare by eye, but there seem to be great similarities (referring to the figures in the appendix). I would be interested whether one kind of IT appears in all season, or another exists only in summer, or only in MAM and SON, or something like that. This then raises the question, if similar/the same clusters appear across seasons, is it necessary to split the seasons for the clustering?

*l 168 and following* The paragraph starting with "Kendall-tau correlations..." seems to be quite interesting but lacks depth. First, I think there should be a new paragraph, i.e. linebreak. Second, the topic is not introduced at all. It would be helpful to mention the teleconnection indices and then explain that you calculated correlations. Also, you mention you use "some" of the indices but do not motivate which ones. Third, the acronyms used in the equations are not introduced, and the dashes as bullet points can be mistaken for minus signs. Most importantly: I am lacking a discussion of what your resulting numbers mean. I.e. does the first equation imply that IT2 in summer occur more often in a positive AMO phase? Is this reasonable/expected or not? Please elaborate on these results more.

*Sect 5*
The intrusions are identified based on temperature thresholds, the conditional probability $P(heat|intrusion)$ can therefore be expected to be extremely high, at least in the area that is covered by the affected air mass.
Your approach adds to this as it shows any area that is impacted. However, since the information shown in Fig 7 is a combination of many intrusion days, it cannot be distinguished between areas that are hot "by definition" just because the air mass is present, and areas that are hot on an intrusion day, but not covered by the intrusion itself. Could you please elaborate on this. One idea could be to show Fig 7 for cell-wise calculation of the impact: $P(heat\ in\ this\ cell|intrusion\ in\ this\ cell)$ in order to see how big the difference is between this direct vs your "also -remote" impact.

Another thing about this section is the way you look at the contribution. I spent a long time thinking about this. My immediate question was why don't you show $P(heat|no\ intrusion)$, as this would be the intuitive measure to compare $P(heat|intrusion)$ to.
I see how your version of the contribution answers the question "In how many percent of heat days was an intrusion present?" and the intuitive conclusion would be "in this many percent of the heat days, an intrusion was present and therefore played a role, i.e. contributed". However, since the heat cannot be the cause of the intrusion, it is difficult to interpret this in terms of causality.
I would definitely ask you to also show $P(heat|no\ intrusion)$. This shows you how often it was hot, despite no intrusion being present. Contrasting $P(heat|intrusion)$ and $P(heat|no\ intrusion)$ by looking at a difference,

relative risk, odds ratio or log-odds ratio will give insight into how the presence of an intrusion increases or decreases the chance of heat. I.e. when the two probabilities are similar, intrusions do not influence heat very much, while when they are different, you can see the effect of the presence of an intrusion.

*l 259* Here you conclude, that Saharan intrusions in the cold months could be similar to subtropical intrusions. To a reader who is not totally familiar with the literature, it is not obvious why and how Saharan intrusions differ from subtropical intrusions. Please make this more clear.

*l 273* Here you argue that the increasing number of intrusions can be explained by thermodynamic changes as the warming of the troposphere. I cannot follow that argument. An increase in intrusions according to your detection algorithm could either be due to an increase in the synoptic situations causing the advection, or due to a general warming of the atmosphere in the WMed region. The former is not necessarily connected to thermodynamic changes, but rather to dynamic changes in a warming climate. The latter is closely tied to thermodynamics and a warming troposphere, but it might be a false trend due to the detection method. If your WMed is generally warmer and the latitudinal gradient weaker, your indicator variable thresholds might be exceeded without having advection from the Sahara present. Please elaborate on this in more detail.

**Minor Comments**

*Throughout the manuscript:*
Please be consistent in your spelling and the usage of the terms for the air mass (Saharan warm air intrusion, warm-air intrusion, warm intrusion, intrusion, air mass, etc.). It will help the readability if you use one clear term throughout the manuscript. Examples, e.g. 50,53,57 or the title of Section 5.
Similarly: Use cross-section (or a precise term of your choosing), as "cut" can be misunderstood (you use cut, vertical section,...).

*l 14* You are referring to changes over what time period? Climatological short(er) term changes?

*l 38* this should be "forest fires in Portugal"

*l 45* gain insight *into*

*l 46* applicable

*l 50*
I don't understand this sentence. These are the aims of the study, no? What do you mean by "objectives to characterise intrusions into the WMed"?

*l 65-66* This was already mentioned very similarly in l. 43 and following.

*l51/69* Please define the "historical period" in terms of years somewhere.

*l 69* Do you mean daily resolution?

*l 73* Daily maximum temperature at what level?

*l 74* Please define the WMed and Euro-Mediterranean region in terms of lon/lat

*l 77* For better readability I suggest:
"We adapt the definition of Sousa et al. (2019), which takes into account..."

*l 81-82* Please reference equations 1 and 2

*l 90* the gas constant of *dry* air

*l 106* "the detected events"

*l 120* Why are you showing August 2006 here and May 2015 in Fig 1? Are they specifically representative? But why chose 2 different ones?

*l 121*
Sardinia is not at 17°E? Do you mean the air mass penetrates *east* of Sardinia?
also: "intense trough in the North Atlantic"

*l 132* "different expressions" is not a precise term, maybe rather use "properties"?

*l 137* I am not sure whether I understand this clustering correctly. You use $\Delta GH$ and $\bar{\bar{\theta}}$ in the WMed region on days that were identified as intrusion days? And then you want to cluster those days into different types. Your results seem as if the position of the anomalously high indicator values defines the clusters. But would the clustering also pick up on two clusters that affect the same region but with different magnitude?

*l 151* Looking at the plot for JJA, I cannot really see how you conclude on 3 clusters. The elbow seems to be higher and the high value in the Silhouette score looks a bit like an irregularity?

*l 151* The last sentence in this paragraph "Finally,..." is really inherent to the clustering method and not necessary here.

*l 155* You are referring to Fig S6, not 5 here.

*l 162* I think this should be DJF and SON, you are referring to panels a,c.

*Fig 6* How do you handle the duration of events, where a long period of intrusion is interrupted by a very short period, where the 5% criterion is just not met for a short time? If you handle these as separate events, this would reduce the mean duration very much compared to when you use a filling or filter.

*l 167* from Fig S8, to me it looks more like IT1 and IT2 often transition to IT3, not IT1 to IT2 and IT3? Or are you interpreting 3 in more than 20 as often?

*l 178* This is not the first time you use the term Euro-Mediterranean. Please introduce the acronym and the exact definition when you mention it first (in the data section?)

*l 180* the 90th percentile of what variable (there is simply a word missing I think).

*l 197* What do you mean with "largest contributions *come from* JJA"?

*l 212* subtracting them *from* the intrusion days' composite

*l 217* "for" instead of "from" in the parentheses

*l 223* What do you mean with "the [...] circulation and SLP anomalies are coherent"?

*l 224* What is the tilde over the 3?

l 225 alway*s*

*l 228* negative sea level pressure *anomaly*

*l 243* Your description earlier implied you are detecting intrusions that originate in the entire Sahara, why are you mentioning only the western Sahara here now?

*l 251* You do not really show that the threshold you choose is robust, or how sensitive result are to it's choice.

*l 256* What is a "climatological air mass"? I understand what you are referring to, but this can be misleading. Also "the latitudinal band" is not precise.

*l 262* Please leave out the "backward", as the studies you cite use forward and backward trajectories.

*l 282* What do you mean by "historical contribution"?

*l 292* remove parenthesis, "which" instead of "and" afterwards

**Comments about Figures**

While many of your figures are nice in general, all have very small font sizes on the axis labels, the axis ticks, etc. Please change this for better readability. Often the use of shared axes or removal of figure titles that are redundant with the caption can create space.

*MAJOR: Fig S1* Caption is wrong, the red box is not the Saharan region. Also here the figure title is redundant. Axis ticks are not necessary in all rows and columns but only at the bottom and left, which makes space for bigger labels ad better readability. Please mention what contours and shading represent in the caption.

*Fig 1:* The green and cyan lines are not so easily distinguishable, please use additional different linestyle. The black dashed box is supposed to be an example for how big 5% are, this needs to be stated very clearly, otherwise it is not obvious that this is just exemplary.

*Fig 2* I do not see the additional value of this figure. When it is mentioned in line 96 it is rather confusing. As far as I understand the helpful information here is that the averages over the Saharan region are in the high tails of the distribution of the WMed region, I think this can be mentioned without showing the figure. One other question this figure raised for me is the spread in the blue dots in February.

*Fig 3* Add dates to the caption when the exact time of the event is, and what you refer to as before, during and after.

*Fig 4* Caption could be clearer. These are composites of the intrusion days of the respective type in the JJA season? The labels in the contour lines are difficult to read and interpret.

*Fig 5* As IT 1-3 are categorical values, a non-sequential colourmap would make the perception easier. Especially in connection with the figures in the Supplement, where more ITs exist. Then the question is also, relating to an earlier comment of mine, whether all IT1 across seasons are similar to each other or not - using the same colour for all IT1 implies that they are more similar to each other than to others.

*Fig 6* Do I understand it correctly, that the orange bar can never be larger than the blue? Consequently, when I only see orange, this means it is a single event?

*Fig 7/8/9* Can you add the area of influence as a contour. You discuss the relation between the impact and the area of influence (e.g. l 192), but it is difficult to see that by comparing differnt plots on different pages.

*Fig 9* be constistent in use of SLP and PSL, I think it should be wind speed *anomalies* in the caption here. And what arrows are you referring to in panels (a) that are not the quivers? This caption is not easy to follow. It might also help to add numbers to the climatological contours for reference. And the quivers are way too small to see their direction properly.
Is Fig 9 panel (a) the same as Fig (4) panel (b) (except for the significance masking)???

---

## Author Comment (AC1)

Note the following errata in the first submission that will be addressed in the revised publication:
In the whole text, the potential temperature indicator is said to be taken as the mean potential temperature between 925 and 750 hPa. We actually used the potential temperature at 700 hPa for the calculations as specified in the literature, the "750" in the text is a typing error. The authors want to apologize in advance.

**General Comments:**

This manuscript is devoted to studying Saharan warm air intrusions in the Western Mediterranean during a historical period (1959-2022), emphasising not only their identification methodology (considering different intrusion types according to their location), and the resulting creation of a catalogue of events, but also the assessment of their impacts on air temperature extremes and the analysis of the large-scale atmospheric driving mechanisms, namely upper-tropospheric anomalies over North Africa. Overall, the topic of research is within the scope of WCD and pertinent under a climate change context and increasing occurrence of temperature extreme events. The datasets used (daily tropospheric variables obtained from the ERA5 reanalysis) are adequate for the purposes of the study, whereas the methodology to identify the different intrusion types provides some novelty. The methodological approach for the detection of Sahara air intrusions in Western Europe is simple and effective, with low computational costs. The results are scientifically sounding, based on sufficient evidence provided by the authors and are in line with previous research outcomes. Generally, the figures are of good quality and the length of the manuscript is adequate. Therefore, I recommend the publication of the manuscript after some revisions that are outlined below in the specific comments.

We want to thank the reviewer for taking the time to read the text and provide feedback that will greatly improve the quality of the text

**Specific Comments:**

Abstract: You mention four ITs. However, depending on the season, different numbers are obtained and considered in the manuscript. Please revise.

Thanks for pointing this out, it will be addressed in the revised version. The number of ITs per season will be specified.

Section 1: There is some unnecessary repetition of citations to previous studies (e.g., Sousa et al. 2019), while other relevant references are still missing (e.g., 10.3402/tellusa.v67.26032, among others). I suggest a more in-depth literature review, as many previous studies have already linked temperature extremes in the Western Mediterranean to atmospheric large-scale mechanisms and anomalies. This is particularly pertinent to corroborate the present study results and provide a good state-of-the-art on this topic of research, giving sufficient credit to other preceding studies.

Thanks for pointing this out. The literature on the link between temperature extremes in the Western Mediterranean and atmospheric large-scale mechanisms and anomalies will be expanded in the introduction.

Ln 50-65: I would avoid bullet-point lists. Providing more compact text is commonly preferable in scientific manuscripts.

Thanks for the suggestion. We considered bullet-point lists to make the objectives and structure of the work more explicit. We would like to keep the objectives as a list to make them more visually available, but we will merge the sections into the text as the reviewer suggests.

Ln 54: please avoid using "etc". Be more specific.

Thanks for pointing this out, it will be changed to "spatial distribution, seasonality and trends"

Section 2: The dataset selected for the analysis was ERA5. Although this choice is adequate for the goals of the study, it should be complemented with weather station data (e.g., from ECA&D), as reanalyses commonly present important biases in near-surface weather conditions, particularly in capturing small-scale spatial variability and local weather/climate patterns. The use of maximum temperatures (TX) recorded at local weather stations would help validate the occurrence of temperature extremes. Hence, I suggest the incorporation of some weather station data to confirm and better quantify the intensity of the extreme events. From my point of view, this is an important limitation on the robustness of the study, though it is not expected a major impact on the results.

We want to thank the reviewer for their comment as adding some observational results will make the study more robust. Therefore, we will include results from some ECA&D stations in the Euro/Mediterranean region. As depicted in the reworked Figure R2-1 (corresponding to the initial submission Figure 7). A table in the supplementary material will be added to document the stations used.

[Figure]

Figure R2-1: Impact of the Saharan warm air intrusions on extreme temperatures in the EM region for the different seasons (rows) and ITs (columns). The impact is measured as the percentage of intrusion days that coincide with an extreme temperature day (TX90p). The amount of intrusion days for each season and IT is specified in the title of each panel. Colored dots are results from ECA&D station data. The area of impact is displayed in green contours, and represents the amount of Saharan air masses recorded in the historical period for a specific IT. The values of the contours go from one Saharan air mass event to the maximum number of days recorded in a gridpoint. Note that the outer contour always represents one recorded day in presence of a Saharan air mass.

Section 3:

- Please do not use physical units in italic font. It will be addressed in the revised text.

- Rd is the "dry air gas constant". Thanks for pointing this out, it will be addressed in the revised text.

- Cp should be cp (lowercase C). Thanks for the suggestion, the text will be modified.

Figure 1: the dashed black box within the Western Mediterranean box is not sufficiently explained in both the caption and text. It seems that the selected day for Fig. 1 does not fulfil the corresponding area requirement.

Thanks for pointing this out. This point will be explained in a better way in the revised text.

Ln 101: "rolling" or "moving"?

In the text we use "rolling" and "moving" (means or windows) interchangeably. We will homogenise all instances to "rolling" to be coherent.

Ln 103 & 104: "red shaped" or "orange"? It will be addressed.

Figure 2 shows a strong positive correlation between both variables. The same is clear in Figure 4. This hints at the high level of redundancy (colinearity) between variables and the information they bring to the analysis. This point should be better discussed, namely its potential implications on the results and the reasoning for their use under the occurrence of temperature extremes driven by Saharan warm air intrusions. For instance, why are you using two variables that deliver similar information and not using variables that complement each other?

The two variables are correlated but not perfectly, and therefore they can complement each other. Using the geopotential thickness and the mean potential temperature aims at identifying air masses with a sufficiently high temperature, and a consistent air mass volume. Furthermore, the potential temperature is more resistant to perturbations, such as Rosby wave propagation, than the geopotential thickness as it should only be affected by diabatic processes. We want to thank the reviewer for this question, and we will expand this reasoning in the text.

Figure 3: the colour scales are not adequate for representing potential temperature (I suggest a rainbow scale or similar). Please revise. Further, colours are not consistent between panels and the description in the caption.

Thanks for pointing this out, the problem we find with the rainbow map is that it is not suited for certain types of colorblindness. We will maintain the Colormaps with python's "inferno" (https://matplotlib.org/stable/gallery/color/colormap_reference.html) coloring if there are no major objections.

Ln 140: please remove ", similar to". It will be addressed in the revised text.

Section 7: I suggest splitting the Discussion and Conclusions section into two separate sections.

Due to the nature of the current work the authors believe the Conclusions and Discussions are more suited together than split into two sections.

Ln 275: IPCC in uppercase. It will be addressed

Ln 292: "as in (Sicard et al., 2022)". Please revise the citation formats throughout the manuscript. Thanks for the suggestion. They will be addressed in the revised text.

Last paragraph of Section 7: the possible linkage of the upper-tropospheric anomalies underlying the onset and establishment of Saharan intrusions to the westerly eddy-driven jet stream over the North Atlantic, including anticyclonic/cyclonic wave-breaking processes, should be discussed.

Thanks for the suggestion, we will expand this discussion in the revised version.

---

## Author Comment (AC2)

Note the following errata in the first submission that will be addressed in the revised publication:
      In the whole text, the potential temperature indicator is said to be taken as the mean potential temperature between 925 and 750 hPa. We actually used the potential temperature at 700 hPa for the calculations as specified in the literature, the "750" in the text is a typing error. The authors want to apologize in advance.

**Review of: "Saharan warm air intrusions in the Western Mediterranean: identification, impacts on temperature extremes and large- scale mechanisms" by Cos et al.**

**Summary and General Assessment**

Warm air from the Sahara can be advected to the Mediterranean and possibly cause extreme temperatures in the Mediterranean and beyond. Cos et al. use an identification method based on the geopotential thickness between 1000 and 500 hPa, and the mean potential temperature between 925 and 750 hPa to investigate the so-called Saharan warm air intrusions into the Western Mediterranean. Using this method, they calculate a catalogue of past intrusions. The authors perform a clustering of intrusion days based on the distribution of geopotential thickness and mean potential temperature, and find three to five distinct clusters per season, that mainly differ in their geographical location. The transition probabilities between the clusters, the correlations with the teleconnection indices, and the composites of meteorological fields are computed per cluster per season. Extreme temperatures are often observed during intrusion days. Based on the conditional probabilities the authors argue that intrusions have a great impact on heat events, but not such a big contribution, i.e. intrusions often cause heat, but heat is not always caused by intrusions. The composites of the meteorological fields show that the different locations of the clusters are due to different locations of the driving pressure patterns.

The study is motivated by increasing temperatures in the western Mediterranean and the mention of such intrusions in the literature. While the connection between the intrusions and heat is not too surprising, the concept of Saharan warm-air intrusions into the western Mediterranean gives a new perspective on extreme temperature events in the region.

The overall structure of the study is well-organised, however some parts need improvement as they are not so easy to follow. The figures support the presented results mostly well, but their readability could be improved. Below I have major, minor, and figure-specific concerns, comments, and questions that should be addressed.

We want to thank the reviewer for taking the time to read the text and provide feedback that we find to greatly improve the quality of the manuscript.

**Major comments**

Section 3

In this section, you describe the algorithm used to identify the intrusions, based on the two indicator variables

$\Delta GH500-1000$ and $\theta750-925$. You refer to the work of Sousa et al. (2019), however, I still lack a motiva- tion/explanation for why these levels are useful to determine the air mass. Especially considering that daily and seasonal variations may change the height of the boundary layer

height drastically, for example. Have you looked at how sensitive the results are to your choice of indicators?

Thanks for this very pertinent comment. We agree that it might not be enough to follow the methodology from Sousa et al. 2019. Their choices come from the climatological characteristics of the Sahara region described in the book "An Introduction to the Meteorology and Climate of the Tropics" by J.F.P. Galvin from 2016.
To verify that the definition can work for our case, we looked at the monthly climatologies of the vertical profiles of the troposphere over the Saharan and Mediterranean. Figure R1 shows the climatological vertical profiles in the Mediterranean region (blue) and in the Sahara (red) for every month of the year.

Summer months have a most distinct lower troposphere potential temperature in the two regions, especially above 950/925 hPa. This is an argument in favour of using 925hPa as the lower bound. From April to October, the potential temperature vertical gradient is larger between 1000hPa and around ~925hPa, then becomes lower from 925hPa to around 700/600hPa. This is the main characteristic of Saharan air masses in the warm months, which are warm and homogeneous in the vertical due to surface longwave radiation heating a thick layer of the lower troposphere over the Sahara.

In colder months, the Saharan and Mediterranean potential temperature vertical gradients are similar. Between November and March, the difference in the vertical profiles lies only in the magnitudes of the potential temperature in all vertical levels, but not in their vertical gradients. Even if the vertical temperature profile The difference in potential temperatures is still relevant to distinguish between Saharan and Mediterranean air masses (as seen in the inspection of the events), although the Saharan air mass is not as homogeneous in the vertical as in the warm months. An intrusion of such an air mass in the cold months should still be distinguishable and have an impact on the Mediterranean region.

In terms of geopotential height, Figure R1 gives little insight as the vertical gradient is much larger than the latitudinal gradient which separates the two regions. We plot Figure R2, which shows the monthly climatological geopotential thickness between different levels. The results suggest that the geopotential thickness between 500 and 1000 hPa has sufficient separation between its Saharan and Mediterranean climatologies. Nonetheless, we perform some sensitivity studies.

[Figure]

Figure R1: Monthly climatologies of vertical potential temperature (solid) and geopotential (dashed) profiles in the Mediterranean region (blue) and in the Sahara (red). The climatology of different months can be seen in the different panels. The shaded area is the interannual variability along the period 1981-2010.

[Figure]

*Figure R2: Climogram with monthly climatologies of geopotential thickness between different levels in the WMed (dashed) and in the Saharan (solid) regions. The shaded area is the interannual variability along the period 1981-2010.*

A sensitivity study of the choice of the levels in the two metrics has been performed using different levels of geopotential thickness ($\Delta$GH500−1000 and $\Delta$GH500−850) and mean potential temperature ($\theta$700−925 and $\theta$700−850). The captured intrusions remain similar.

In summer, similar results are obtained for the combination of indicators $\theta$700-925/$\Delta$GH500-1000, $\theta$700-850/$\Delta$GH500-1000 and $\theta$700-925/$\Delta$GH500-1000 (The last one does not capture Sousa et al. 2019 events). $\theta$700-850/g500-850 identifies longer events and exceeds the intrusion day classification from Sousa et al. 2019. In winter the choice of metric is not sensitive to the levels used. In spring and autumn, changes in the levels used have some small influence in the length of the events.

In general, using a higher lower potential temperature bound leads to more and lengthier events, and using a lower geopotential bound leads to a decrease in the amount and length of the events.

We will expand the sensitivity of the study on the levels in the revised version hoping to obtain a more robust manuscript.

[Figure]

*Figure R3: Catalogues of intrusion days in winter (DJF) from 1959/1960 to 2021/2022 using 4 different indicator combinations: i) θ700-850/ΔGH500-850, ii) θ700-850/ΔGH500-1000, iii) θ700-925/ΔGH500-850 and iv) θ700-925/ΔGH500-1000 (from left to right). The days with intrusion are painted in dark red, and the panels below the catalogues show the aggregated number of intrusion days and mean event duration during DJF for all the period. A least squares linear regression fit to the seasonal number of intrusion days is shown in blue.*

[Figure]

*Figure R4: Same as R3 but for MAM in the period 1959 to 2022.*

[Figure]

*Figure R5: Same as R3 but for JJA in the period 1959 to 2022.*

[Figure]

*Figure R6: Same as R3 but for SON in the period 1959 to 2022.*

l 91-105

This paragraph is very difficult to follow. It can be improved by rewriting/rephrasing, but since it is a crucial paragraph to the paper, I consider this a major comment. Let me summarise what I understood:

Monthly climatologies of ΔGH and θ show that in summer values are maximised over the Sahara, but during the rest of the year they decrease slowly and constantly between 15 and 35∘N.

Then, you choose a region as representative for the Sahara. You average over this region, which was chosen so that you meet the values described in literature. I am missing the exact extent of this region in the text.

Then, you calculate 31-day rolling means, averaged over the region, to define thresholds. Why do you chose 31d in particular and how sensitive is your result to this choice?

The air mass is then defined as all grid cells that exceed both threshold values. Does this always yield a coherent air mass as depicted in Fig 1? I would imagine there might be single cells within the air mass that are not identified, or single cells outside of it that are?

Please improve this paragraph by explaining step by step what you did and why.
In lines 109-115 you mention again that in the colder seasons, the values in the Sahara are not as distinct, since there is a constant gradient present. First, this is a little bit repetitive to what was mentioned before (l 92 following). Second, you do not really explain what the consequences of this are. I would imagine that the sensitivity to the threshold values varies a lot across seasons because of this? Does this mean your method works better in summer than in the rest of the year?

We see how this paragraph might be confusing and understand that it raises questions about sensitivity to parameter choice. Therefore, sensitivity studies have been conducted (Figures R1-R6) in order to better illustrate the implications of changing the pressure levels where metrics are computed. In the revised text we try to explain better the Sahara climatological properties, taking advantage of the figures that were produced thanks to Reviewer #1's previous question.

**Answers to other questions:**

Why using a 31-day rolling mean?
The original work by Sousa et al. 2019 uses the climatology of the summer months as thresholds for the whole summer. We identified that using the climatology of three months for the thresholds makes the identification criteria vary for events along the 3-month period. For instance, using the 3-month JJA climatology might make early June and late august events harder to identify than events in the middle of summer (closer to the climatological maximum temperature). Therefore we decided to make a rolling climatology that would take into account the seasonal cycle of the metrics. Different periods were explored, from 15-day to 3-month rolling means. The choice of 15 days was very restrictive, while three months was too permissive. After exploring the results we went for a 31-day rolling mean, which although arbitrary, is an educated guess.

Spatial continuity of the intrusions
We see, from the visual inspection of all the events, that there are no spurious masses generated outside the Sahara. From this inspection we could ascertain that during Saharan warm air intrusions, air masses are linked to air mass movements from the Sahara to the Mediterranean. Nevertheless, there are intrusion days when the Saharan air mass seem to split up, most likely due to orographic effects.

Consequences of the winter less distinct Saharan properties
The consequences of having less distinct values in the Sahara during the cold months is that air masses that are considered to be within the threshold of a Saharan air mass might not come directly from the Sahara, but maybe from the Atlantic ocean at Saharan latitudes. The properties of these air masses are still uncommon in the Mediterranean region in the winter season. That is why some intrusions might be more precisely referred to as subtropical ridges in this season.

We have modified the text so that the differences between seasons are better understood and the thresholds used fully justified.

l 107

"air masses that are displaced from the Sahara". Does that mean the air mass has to cover parts of the Sahara at all times to ensure the origin?
Is the visual inspection necessary for your detection algorithm in general, or was this just a proof of concept in this case? Because if there is doubt about the origin of the air masses, and this step is always necessary, this is a big issue in the applicability of the method to other cases.

The inspection was a proof of concept to make sure that the parameters chosen make physical sense. Also, as a method based on thresholds with a certain degree of arbitrariness, some parameter tuning is needed to make sure that the method is actually capturing real events and avoid spurious warm air masses generated away from the Sahara.
The only issue that can't be detected from the visual inspection is the difficulty to detect events that are not captured by the method. This is a consequence of the novelty of this work, which makes that an alternative method to detect missed events is not available yet. This will be discussed in the revised version of the paper.

l 116-118

As far as I understand, the development of the catalogue and analysis of seasonal and inter-annual variations is a crucial part of your study. I was therefore disappointed to be referred to the Supplement for such a central information. Why don't you show an average number of events per month and their spread across years (e.g. as a boxplot). Then your description of the seasonal variability can refer to this plot.

[Figure]

[Figure]

*Figure R7: Number of identified Saharan warm air intrusions in a) DJF, b) MAM, c) JJA and d) SON per year between 1959-2022 (blue bars). The mean duration of each intrusion event, from the first to the last continuous day, is shown with orange bars. Note that when a single event is recorded for a specific year the blue and orange bars are the same size, therefore, only the orange is shown. A least squares linear regression fit to the seasonal number of intrusion days is shown in blue.The panels on the right show the distribution of intrusion days for each month within the season and the whole season.*

We want to thank the reviewer for pointing this out. We agree that leaving the catalogue in the supplementary material goes against the interests of this publication and we will produce the suggested plot displaying the seasonality of events in the revised version of the paper. We will move Figure 6 earlier in the text and add visual data that informs about the average intrusion days per month and its interannual variability. An example is presented in Figure R7

Fig 3
From the cross-sections it becomes obvious that the air mass tilts up towards the edges, especially in the North and West. This will not be captured by your methodology, which focuses on the lower levels only. I think, however, this warm, dry air mass might still have considerable effects in those regions. This goes in the same direction as the question about how sensitive your results are to the choice of the indicator variables and their respective levels. Could you discuss this a little bit further?

The effects of a warm air mass in the higher parts of the troposphere are not direct but rather a consequence of the air mass intrusion on the atmospheric dynamics.  This indirect effect

escapes the scope of the current work. Further studies with the catalogue will focus on the extratropical dynamical effects and the impacts on cloud cover and surface downward shortwave radiation.

I 153 and following and Fig 4
I was wondering how similar the clusters between the seasons are. The different colour scales make it difficult to compare by eye, but there seem to be great similarities (referring to the figures in the appendix). I would be interested whether one kind of IT appears in all season, or another exists only in summer, or only in MAM and SON, or something like that. This then raises the question, if similar/the same clusters appear across seasons, is it necessary to split the seasons for the clustering?

The clusters look similar, but we prefer to separate the clusters per season because we think that it is interesting to illustrate the large-scale characteristics across seasons. The large-scale structure shows differences, therefore we decided to keep the results of the clustering separated. We will make this motivation more explicit in the text

I 168 and following The paragraph starting with "Kendall-tau correlations..." seems to be quite interesting but lacks depth. First, I think there should be a new paragraph, i.e. linebreak. Second, the topic is not introduced at all. It would be helpful to mention the teleconnection indices and then explain that you calculated correlations. Also, you mention you use "some" of the indices but do not motivate which ones. Third, the acronyms used in the equations are not introduced, and the dashes as bullet points can be mistaken for minus signs. Most importantly: I am lacking a discussion of what your resulting numbers mean. I.e. does the first equation imply that IT2 in summer occur more often in a positive AMO phase? Is this reasonable/expected or not? Please elaborate on these results more.

We agree that the results and discussion around the teleconnection indices is shallow. Therefore, we will expand on this and justify the different indices used with the literature of Mediterranean drivers. Also, we will discuss the implications of obtaining significant correlations with different teleconnection indices.

Sect 5
The intrusions are identified based on temperature thresholds, the conditional probability P (heat|intrusion) can therefore be expected to be extremely high, at least in the area that is covered by the affected air mass.
Your approach adds to this as it shows any area that is impacted. However, since the information shown in Fig 7 is a combination of many intrusion days, it cannot be distinguished between areas that are hot "by definition" just because the air mass is present, and areas that are hot on an intrusion day, but not covered by the intrusion itself. Could you please elaborate on this. One idea could be to show Fig 7 for cell-wise calculation of the impact: P (heat in this cell|intrusion in this cell) in order to see how big the difference is between this direct vs your "also -remote" impact.

Another thing about this section is the way you look at the contribution. I spent a long time thinking about this. My immediate question was why don't you show P (heat|no intrusion), as this would be the intuitive measure to compare P (heat|intrusion) to.

I see how your version of the contribution answers the question "In how many percent of heat days was an intrusion present?" and the intuitive conclusion would be "in this many percent of the heat days, an intrusion was present and therefore played a role, i.e. contributed". However, since the heat cannot be the cause of the intrusion, it is difficult to interpret this in terms of causality.

I would definitely ask you to also show P (heat|no intrusion). This shows you how often it was hot, despite no intrusion being present. Contrasting P (heat|intrusion) and P (heat|no intrusion) by looking at a difference, relative risk, odds ratio or log-odds ratio will give insight into how the presence of an intrusion increases or decreases the chance of heat. I.e. when the two probabilities are similar, intrusions do not influence heat very much, while when they are different, you can see the effect of the presence of an intrusion.

[Figure]

*Figure R8: Odds ratio of the Impact of the Saharan warm air intrusions on extreme temperatures in the EM region for the different seasons (rows) and ITs (columns). The odds ratio is measured as the fraction of the percentage of intrusion days that coincide with an extreme temperature day (TX90p) divided by the percentage of extreme temperature days that do not coincide with an intrusion day. The area of impact is displayed in green contours, and represents the amount*

*of Saharan air masses recorded in the historical period for a specific IT. The values of the contours go from one Saharan air mass event to the maximum number of days recorded in a gridpoint. Note that the outer contour always represents one recorded day in presence of a Saharan air mass.*

We thank very much this thorough comment on the role of the intrusions to extreme temperatures. We will add contour lines showing which is the accumulated extent of intrusion days for each IT (effectively showing the area of impact, which will be redefined to the boundary where intrusions have been recorded).
Regarding the probabilities to show, it was a tricky part to convey and could definitely be improved by using the P(heat|no intrusion). We computed the odds ratio and displayed it in Figure R8. The regions where the probability of having an extreme is higher when intrusions happen are shaded in red. These results will be added to the revised manuscript and discussed.

l 259 Here you conclude, that Saharan intrusions in the cold months could be similar to subtropical intrusions. To a reader who is not totally familiar with the literature, it is not obvious why and how Saharan intrusions differ from subtropical intrusions. Please make this more clear.

Thanks for pointing this out. We will expand on this so that the difference between a Saharan intrusion and a subtropical intrusion is better understood. We believe that introducing this concept when we talk about the Sahara climatologies in the cold months might be helpful.

l 273 Here you argue that the increasing number of intrusions can be explained by thermodynamic changes as the warming of the troposphere. I cannot follow that argument. An increase in intrusions according to your detection algorithm could either be due to an increase in the synoptic situations causing the advection, or due to a general warming of the atmosphere in the WMed region. The former is not necessarily connected to thermodynamic changes, but rather to dynamic changes in a warming climate. The latter is closely tied to thermodynamics and a warming troposphere, but it might be a false trend due to the detection method. If your WMed is generally warmer and the latitudinal gradient weaker, your indicator variable thresholds might be exceeded without having advection from the Sahara present. Please elaborate on this in more detail.

We agree with the reviewer, the increase could also be due to a change in the dynamics induced by climate change. We will include the discussion of how climate change might also affect changes in the dynamics and in turn have an effect on Saharan warm air intrusions.
Regarding the increasingly WMed region and the indicator thresholds being more easily exceeded, we agree that there might be intrusions that correspond to an advection of air masses from the north of the Sahara region. This could be the case in the most recent years due to warming. Nonetheless, the characteristics of the air mass will be climatologically Saharan. There is literature indicating projected desertification of Northern Africa, Southern Iberian Peninsula and other regions of the Southern Mediterranean (lite lite lite). If this is the case, we shouldn't be surprised about Saharan-like air masses being formed north of the

Sahara in a warming climate. The framework we present here, aims to define a Saharan air mass in the recent historical period to be used as a baseline (benchmark).

We thank the reviewer for the thorough minor comments that follow, they will greatly improve the text. We will apply the suggestions to the text. See specific comments that require answers below:

**Minor Comments**

Throughout the manuscript:

Please be consistent in your spelling and the usage of the terms for the air mass (Saharan warm air intrusion, warm-air intrusion, warm intrusion, intrusion, air mass, etc.). It will help the readability if you use one clear term throughout the manuscript. Examples, e.g. 50,53,57 or the title of Section 5.

Similarly: Use cross-section (or a precise term of your choosing), as "cut" can be misunderstood (you use cut, vertical section,...).

l 14 You are referring to changes over what time period? Climatological short(er) term changes? We refer to both historical and projected. We will specify in the text.

l 38 this should be "forest fires in Portugal"

l 45 gain insight into

l 46 applicable

l 50 I don't understand this sentence. These are the aims of the study, no? What do you mean by "objectives to characterise intrusions into the WMed"? Agreed, sentence is changed to "This study has four objectives with regard to Saharan warm air intrusions (referred to in the text as SWAIs) into the Western Mediterranean …:"

l 65-66 This was already mentioned very similarly in l. 43 and following. The last sentence is removed.

l51/69 Please define the "historical period" in terms of years somewhere.

l 69 Do you mean daily resolution?

l 73 Daily maximum temperature at what level?

l 74 Please define the WMed and Euro-Mediterranean region in terms of lon/lat

l 77 For better readability I suggest:

"We adapt the definition of Sousa et al. (2019), which takes into account..."

l 81-82 Please reference equations 1 and 2

l 90 the gas constant of dry air

l 106 "the detected events"

l 120 Why are you showing August 2006 here and May 2015 in Fig 1? Are they specifically representative? But why chose 2 different ones? There is no specific reason for choosing one or the other. Both correspond to intrusion days. We think it is good to show different events.

l 121 Sardinia is not at 17∘E? Do you mean the air mass penetrates east of Sardinia? also: "intense trough in the North Atlantic". That is right, wrong synaptic connection from my end. I meant Sicily, it will be corrected.

l 132 "different expressions" is not a precise term, maybe rather use "properties"?

l 137 I am not sure whether I understand this clustering correctly. You use ∆GH and θ in the WMed region on days that were identified as intrusion days? And then you want to cluster those days into different types. Your results seem as if the position of the anomalously high indicator

values defines the clusters. But would the clustering also pick up on two clusters that affect the same region but with different magnitude? From the obtained results we gave more weight to the interpretation of the positioning of the intrusion, but it is true that the clustering seems to also capture differences in the magnitude (IT2 of Figure 4 clearly has larger magnitude as well as other ITs in the rest of seasons). Thanks for the comment, we will include these findings in the results and discussion.

l 151 Looking at the plot for JJA, I cannot really see how you conclude on 3 clusters. The elbow seems to be higher and the high value in the Silhouette score looks a bit like an irregularity?

l 151 The last sentence in this paragraph "Finally,..." is really inherent to the clustering method and not necessary here.

l 155 You are referring to Fig S6, not 5 here.

l 162 I think this should be DJF and SON, you are referring to panels a,c. It is DJF, JJA and SON which are panels a,c,d. It will be fixed

Fig 6 How do you handle the duration of events, where a long period of intrusion is interrupted by a very short period, where the 5% criterion is just not met for a short time? If you handle these as separate events, this would reduce the mean duration very much compared to when you use a filling or filter. We are considering them as separate events.

l 167 from Fig S8, to me it looks more like IT1 and IT2 often transition to IT3, not IT1 to IT2 and IT3? Or are you interpreting 3 in more than 20 as often? That is right, IT1 and IT2 often transition to IT3. It will be addressed in the textl 178 This is not the first time you use the term Euro-Mediterranean. Please introduce the acronym and the exact definition when you mention it first (in the data section?)

l 180 the 90th percentile of what variable (there is simply a word missing I think).

l 197 What do you mean with "largest contributions come from JJA"? The largest contribution of extreme temperatures that come from intrusion days (P(intrusion day|TX90)).

l 212 subtracting them from the intrusion days' composite

l 217 "for" instead of "from" in the parentheses

l 223 What do you mean with "the [...] circulation and SLP anomalies are coherent"? That the wind directions and SLP fields in the composites make sense, which shouldn't be surprising. It will be made clearer in the text.

l 224 What is the tilde over the 3? It is a typographic error, thanks for noticing.

l 225 always

l 228 negative sea level pressure anomaly

l 243 Your description earlier implied you are detecting intrusions that originate in the entire Sahara, why are you mentioning only the western Sahara here now? This is a typing error it should say only Sahara

l 251 You do not really show that the threshold you choose is robust, or how sensitive result are to it's choice. We will avoid saying robust.

l 256 What is a "climatological air mass"? I understand what you are referring to, but this can be misleading. Also "the latitudinal band" is not precise. This will be re-worked in line with the second major comment

l 262 Please leave out the "backward", as the studies you cite use forward and backward trajectories.

What do you mean by "historical contribution"?

l 292 remove parenthesis, "which" instead of "and" afterwards

We want to thank the suggestions done in the Figures. We will apply them in the revised version as we agree with all the suggestions

**Comments about Figures**

While many of your figures are nice in general, all have very small font sizes on the axis labels, the axis ticks, etc. Please change this for better readability. Often the use of shared axes or removal of figure titles that are redundant with the caption can create space.

MAJOR: Fig S1 Caption is wrong, the red box is not the Saharan region. Also here the figure title is redundant. Axis ticks are not necessary in all rows and columns but only at the bottom and left, which makes space for bigger labels ad better readability. Please mention what contours and shading represent in the caption.

Fig 1: The green and cyan lines are not so easily distinguishable, please use additional different linestyle. The black dashed box is supposed to be an example for how big 5% are, this needs to be stated very clearly, otherwise it is not obvious that this is just exemplary.

Fig 2 I do not see the additional value of this figure. When it is mentioned in line 96 it is rather confus- ing. As far as I understand the helpful information here is that the averages over the Saharan region are in the high tails of the distribution of the WMed region, I think this can be mentioned without showing the figure. One other question this figure raised for me is the spread in the blue dots in February.

We will consider moving this figure to the supplement and using information from Figures R1 and R2 to better convey our point.

Fig 3 Add dates to the caption when the exact time of the event is, and what you refer to as before, dur- ing and after.

Fig 4 Caption could be clearer. These are composites of the intrusion days of the respective type in the JJA season? The labels in the contour lines are difficult to read and interpret.

Fig 5 As IT 1-3 are categorical values, a non-sequential colourmap would make the perception easier. Espe- cially in connection with the figures in the Supplement, where more ITs exist. Then the question is also, relating to an earlier comment of mine, whether all IT1 across seasons are similar to each other or not - using the same colour for all IT1 implies that they are more similar to each other than to others.

Fig 6 Do I understand it correctly, that the orange bar can never be larger than the blue? Consequently, when I only see orange, this means it is a single event? Exactly, when only orange appears means that only one event exists for that season and year. I'll mention it in the caption.

Fig 7/8/9 Can you add the area of influence as a contour. You discuss the relation between the impact and the area of influence (e.g. l 192), but it is difficult to see that by comparing differnt plots on different pages. This is a very good idea, we will add the contours of the area of impact. (See Figure R8)

Fig 9 be constistent in use of SLP and PSL, I think it should be wind speed anomalies in the caption here. And what arrows are you referring to in panels (a) that are not the quivers? This caption is not easy to follow. It might also help to add numbers to the climatological contours for reference. And the quivers are way too small to see their direction properly.

Is Fig 9 panel (a) the same as Fig (4) panel (b) (except for the significance masking)??? Fig 9 (a) and Fig 4 (b) are not the same. Any similarities are due to compositing the same days, although the variables are different

---

## Author Response (AR1)

The authors would like to sincerely thank the comments and suggestions provided by the reviewers as we think they have greatly improved the quality of the study and the text. This round of revisions has, in *dark blue and italic font*, parts of answers from the first Author Comment (the previous round of revision answers) that are kept here for context. In light blue and under/after each comment we answer and explain the changes performed in the revised manuscript. Please find attached the tracked changes file, revised manuscript, and supplement.

REVIEWER #1 COMMENTS

**Review of: "Saharan warm air intrusions in the Western Mediterranean: identification, impacts on temperature extremes and large- scale mechanisms" by Cos et al.**

**Summary and General Assessment**

Warm air from the Sahara can be advected to the Mediterranean and possibly cause extreme temperatures in the Mediterranean and beyond. Cos et al. use an identification method based on the geopotential thickness between 1000 and 500 hPa, and the mean potential temperature between 925 and 750 hPa to investigate the so-called Saharan warm air intrusions into the Western Mediterranean. Using this method, they calculate a catalogue of past intrusions. The authors perform a clustering of intrusion days based on the distribution of geopotential thickness and mean potential temperature, and find three to five distinct clusters per season, that mainly differ in their geographical location. The transition probabilities between the clusters, the correlations with the teleconnection indices, and the composites of meteorological fields are computed per cluster per season. Extreme temperatures are often observed during intrusion days. Based on the conditional probabilities the authors argue that intrusions have a great impact on heat events, but not such a big contribution, i.e. intrusions often cause heat, but heat is not always caused by intrusions. The composites of the meteorological fields show that the different locations of the clusters are due to different locations of the driving pressure patterns.

The study is motivated by increasing temperatures in the western Mediterranean and the mention of such intrusions in the literature. While the connection between the intrusions and heat is not too surprising, the concept of Saharan warm-air intrusions into the western Mediterranean gives a new perspective on extreme temperature events in the region.

The overall structure of the study is well-organised, however some parts need improvement as they are not so easy to follow. The figures support the presented results mostly well, but their readability could be improved. Below I have major, minor, and figure-specific concerns, comments, and questions that should be addressed.

We want to thank the reviewer for taking the time to read the text and provide feedback that we find to greatly improve the quality of the manuscript.

**Major comments**

Section 3

In this section, you describe the algorithm used to identify the intrusions, based on the two indicator variables
$\Delta GH_{500-1000}$ and $\theta_{750-925}$. You refer to the work of Sousa et al. (2019), however, I still lack a motiva- tion/explanation for why these levels are useful to determine the air mass. Especially considering that daily and seasonal variations may change the height of the boundary layer height drastically, for example. Have you looked at how sensitive the results are to your choice of indicators?

To solve this comment, we have added more context and an assessment of the climatological values of the variables (avg{θ} and ΔGH at different levels) in the WMed and Sahara. We also added the Figures presented in the first Author Comment (the previous round of revision answers), which are now Figures S1 and S2 in the supplement of the revised manuscript. Some sensitivity studies have also been added to Section 3 in the revised manuscript.

"The sensitivity to the levels used to define the indicator thresholds is evaluated for the lower bounds of the geopotential thickness ($\Delta GH_{500-1000}$ and $\Delta GH_{500-850}$) and mean potential temperature ($\theta_{700-925}$ and $\theta_{700-850}$). In general, using a higher lower-level potential temperature bound leads to more and lengthier events, and a higher lower-level geopotential bound leads to a decrease in the amount and length of the events. Although some changes appear in the amount of days recorded as intrusions (not shown), the captured intrusions remain similar and long events are always captured with some slight changes in the duration. If we study the impacts in extreme temperatures and the atmospheric circulation associated with intrusion events, the results and conclusions remain the same as the ones conveyed in the subsequent sections of this work."

Find below the answer we gave in the Author Comment for context:

*Thanks for this very pertinent comment. We agree that it might not be enough to follow the methodology from Sousa et al. 2019. Their choices come from the climatological characteristics of the Sahara region described in the book "An Introduction to the Meteorology and Climate of the Tropics" by J.F.P. Galvin from 2016.*
*To verify that the definition can work for our case, we looked at the monthly climatologies of the vertical profiles of the troposphere over the Saharan and Mediterranean. Figure R1 shows the climatological vertical profiles in the Mediterranean region (blue) and in the Sahara (red) for every month of the year.*
*Summer months have a most distinct lower troposphere potential temperature in the two regions, especially above 950/925 hPa. This is an argument in favour of using 925hPa as the lower bound. From April to October, the potential temperature vertical gradient is larger between 1000hPa and around ~925hPa, then becomes lower from 925hPa to around 700/600hPa. This is the main characteristic of Saharan air masses in the warm months, which are warm and homogeneous in the vertical due to surface longwave radiation heating a thick layer of the lower troposphere over the Sahara.*
*In colder months, the Saharan and Mediterranean potential temperature vertical gradients are similar. Between November and March, the difference in the vertical profiles lies*

*only in the magnitudes of the potential temperature in all vertical levels, but not in their vertical gradients. Even if the vertical temperature profile The difference in potential temperatures is still relevant to distinguish between Saharan and Mediterranean air masses (as seen in the inspection of the events), although the Saharan air mass is not as homogeneous in the vertical as in the warm months. An intrusion of such an air mass in the cold months should still be distinguishable and have an impact on the Mediterranean region.*

*In terms of geopotential height, Figure R1 gives little insight as the vertical gradient is much larger than the latitudinal gradient which separates the two regions. We plot Figure R2, which shows the monthly climatological geopotential thickness between different levels. The results suggest that the geopotential thickness between 500 and 1000 hPa has sufficient separation between its Saharan and Mediterranean climatologies. Nonetheless, we perform some sensitivity studies.*

[Figure]

*Figure R1: Monthly climatologies of vertical potential temperature (solid) and geopotential (dashed) profiles in the Mediterranean region (blue) and in the Sahara (red). The climatology of different months can be seen in the different panels. The shaded area is the interannual variability along the period 1981-2010.*

[Figure]

*Figure R2: Climogram with monthly climatologies of geopotential thickness between different levels in the WMed (dashed) and in the Saharan (solid) regions. The shaded area is the interannual variability along the period 1981-2010.*

*A sensitivity study of the choice of the levels in the two metrics has been performed using different levels of geopotential thickness (△GH500−1000 and △GH500−850) and mean potential temperature (θ700−925 and θ700−850). The captured intrusions remain similar.*

*In summer, similar results are obtained for the combination of indicators θ700-925/△GH500-1000, θ700-850/△GH500-1000 and θ700-925/△GH500-1000 (The last one does not capture Sousa et al. 2019 events). θ700-850/g500-850 identifies longer events and exceeds the intrusion day classification from Sousa et al. 2019. In winter the choice of metric is not sensitive to the levels used. In spring and autumn, changes in the levels used have some small influence in the length of the events.*

In general, using a higher lower potential temperature bound leads to more and lengthier events, and using a lower geopotential bound leads to a decrease in the amount and length of the events.

We will expand the sensitivity of the study on the levels in the revised version hoping to obtain a more robust manuscript.

[Figure]

*Figure R3: Catalogues of intrusion days in winter (DJF) from 1959/1960 to 2021/2022 using 4 different indicator combinations: i) θ700-850/ΔGH500-850, ii) θ700-850/ΔGH500-1000, iii) θ700-925/ΔGH500-850 and iv) θ700-925/ΔGH500-1000 (from left to right). The days with intrusion are painted in dark red, and the panels below the catalogues show the aggregated number of intrusion days and mean event duration during DJF for all the period. A least squares linear regression fit to the seasonal number of intrusion days is shown in blue.*

[Figure]

*Figure R4: Same as R3 but for MAM in the period 1959 to 2022.*

[Figure]

*Figure R5: Same as R3 but for JJA in the period 1959 to 2022.*

[Figure]

*Figure R6: Same as R3 but for SON in the period 1959 to 2022.*

I 91-105

This paragraph is very difficult to follow. It can be improved by rewriting/rephrasing, but since it is a crucial paragraph to the paper, I consider this a major comment. Let me summarise what I understood:

Monthly climatologies of ΔGH and θ show that in summer values are maximised over the Sahara, but during the rest of the year they decrease slowly and constantly between 15 and 35◦N.

Then, you choose a region as representative for the Sahara. You average over this region, which was chosen so that you meet the values described in literature. I am missing the exact extent of this region in the text.

Then, you calculate 31-day rolling means, averaged over the region, to define thresholds. Why do you chose 31d in particular and how sensitive is your result to this choice?

The air mass is then defined as all grid cells that exceed both threshold values. Does this always yield a coherent air mass as depicted in Fig 1? I would imagine there might be single cells within the air mass that are not identified, or single cells outside of it that are?

Please improve this paragraph by explaining step by step what you did and why.
In lines 109-115 you mention again that in the colder seasons, the values in the Sahara are not as distinct, since there is a constant gradient present. First, this is a little bit repetitive to what was mentioned before (l 92 following). Second, you do not really explain what the consequences of this are. I would imagine that the sensitivity to the threshold values varies a lot across seasons because of this? Does this mean your method works better in summer than in the rest of the year?

We have rephrased the paragraph for clarity,adding better context of the climatology of the threshold variables (ls. 111-119):

"By conducting a study of the monthly climatologies of the two indicators defined in Equations 1 and 2, we find that the air masses over the Sahara desert behave differently in the warm and cold months (see Figure S3). The indicators in the Sahara region during the warm months have distinct (maximum) values compared to any surrounding region (the Atlantic, the Mediterranean and the equatorial band show lower values). During the cold season, the climatological $\Delta GH500\text{-}1000$ and $\theta 700\text{-}925$, are not as distinct in the Sahara region, but rather have an almost constant latitudinal gradient in the longitudinal band 15ºN-35ºN,30ºW-50ºE (e.g. in winter, the East Atlantic at the same latitudes as the Sahara has comparable $\theta 700\text{-}925$, and $\Delta GH500\text{-}1000$ values). We argue that the lesser radiative forcing in the cold months and the shorter days do not allow the desert areas to heat the troposphere above them as notably as in the warm months, however, the indicator magnitudes over the Sahara are still distinct from those in the WMed region."

We also commented the sensitivities to taking a 31-day rolling window for the thresholds (ls. 129-131)

Finally we have expanded on the differences between detected intrusions in the cold and warm months (ls. 149-155):

"It is noteworthy that, in line with the results obtained for the differing Saharan characteristics in the climatologies of the warm and cold months, intrusions are qualitatively different in May-October than in November-March. In general, during the warm months, narrow bands of Saharan air are advected northward while in the cold months intrusions tend to span a broader longitudinal band that moves in a wavelike pattern with the crest of Saharan warm air reaching the WMed. Note that, as mentioned in previous paragraphs, in the cold months, warm air advections might have mixed origins between the East Atlantic (below ~30 ºN) and the Sahara, but always with Saharan-like warm air properties."

Below we leave the first Author Comment to the revision for context:

*We see how this paragraph might be confusing and understand that it raises questions about sensitivity to parameter choice. Therefore, sensitivity studies have been conducted (Figures R1-R6) in order to better illustrate the implications of changing the pressure levels where*

*metrics are computed. In the revised text we try to explain better the Sahara climatological properties, taking advantage of the figures that were produced thanks to Reviewer #1's previous question.*

**Answers to other questions:**

*Why using a 31-day rolling mean?*
*The original work by Sousa et al. 2019 uses the climatology of the summer months as thresholds for the whole summer. We identified that using the climatology of three months for the thresholds makes the identification criteria vary for events along the 3-month period. For instance, using the 3-month JJA climatology might make early June and late august events harder to identify than events in the middle of summer (closer to the climatological maximum temperature). Therefore we decided to make a rolling climatology that would take into account the seasonal cycle of the metrics. Different periods were explored, from 15-day to 3-month rolling means. The choice of 15 days was very restrictive, while three months was too permissive. After exploring the results we went for a 31-day rolling mean, which although arbitrary, is an educated guess.*

*Spatial continuity of the intrusions*
*We see, from the visual inspection of all the events, that there are no spurious masses generated outside the Sahara. From this inspection we could ascertain that during Saharan warm air intrusions, air masses are linked to air mass movements from the Sahara to the Mediterranean. Nevertheless, there are intrusion days when the Saharan air mass seem to split up, most likely due to orographic effects.*

*Consequences of the winter less distinct Saharan properties*
*The consequences of having less distinct values in the Sahara during the cold months is that air masses that are considered to be within the threshold of a Saharan air mass might not come directly from the Sahara, but maybe from the Atlantic ocean at Saharan latitudes. The properties of these air masses are still uncommon in the Mediterranean region in the winter season. That is why some intrusions might be more precisely referred to as subtropical ridges in this season.*

l 107
"air masses that are displaced from the Sahara". Does that mean the air mass has to cover parts of the Sahara at all times to ensure the origin?
Is the visual inspection necessary for your detection algorithm in general, or was this just a proof of concept in this case? Because if there is doubt about the origin of the air masses, and this step is always necessary, this is a big issue in the applicability of the method to other cases.

As mentioned in the first Author Comment, the inspection helps us tune the algorithm and hence it can now be used for other applications such as Climate Model evaluation. We have added a couple of sentences justifying the visual inspection:

"During the aforementioned sensitivity studies (of the levels used for the indicators and the rolling window climatologies) we performed a visual inspection of all the event days to help in the definition of the algorithm parameters and make sure that the detected events are not spurious air masses formed away from the Sahara. All Saharan warm air intrusion events recorded with the algorithm represent continuous air masses that are displaced northward."

Below the first answer to the review:

*The inspection was a proof of concept to make sure that the parameters chosen make physical sense. Also, as a method based on thresholds with a certain degree of arbitrariness, some parameter tuning is needed to make sure that the method is actually capturing real events and avoid spurious warm air masses generated away from the Sahara.*
*The only issue that can't be detected from the visual inspection is the difficulty to detect events that are not captured by the method. This is a consequence of the novelty of this work, which makes that an alternative method to detect missed events is not available yet. This will be discussed in the revised version of the paper.*

l 116-118
As far as I understand, the development of the catalogue and analysis of seasonal and inter-annual variations is a crucial part of your study. I was therefore disappointed to be referred to the Supplement for such a central information. Why don't you show an average number of events per month and their spread across years (e.g. as a boxplot). Then your description of the seasonal variability can refer to this plot.

We want to thank the reviewer for pointing this out. We have produced the suggested plot displaying the seasonality of events for the revised version of the manuscript. We have moved earlier in the text so now is Figure 2. An example is presented in Figure R7

[Figure]

*Figure R7: Number of identified Saharan warm air intrusions in a) DJF, b) MAM, c) JJA and d) SON per year between 1959-2022 (blue bars). The mean duration of each intrusion event, from the first to the last continuous day, is shown with orange bars. Note that when a single event is recorded for a specific year the blue and orange bars are the same size, therefore, only the orange is shown. A least squares linear regression fit to the seasonal number of intrusion days is shown in blue. e-h) show the boxplots of the annual intrusion days for each month and season.*

Fig 3

From the cross-sections it becomes obvious that the air mass tilts up towards the edges, especially in the North and West. This will not be captured by your methodology, which focuses

on the lower levels only. I think, however, this warm, dry air mass might still have considerable effects in those regions. This goes in the same direction as the question about how sensitive your results are to the choice of the indicator variables and their respective levels. Could you discuss this a little bit further?

Thank you for bringing this up. There might be some indirect effect on surface temperature from having a Saharan-like air mass in the higher troposphere. We already mentioned, in the previous Author Comment, that studying what effect such a tilt can have falls out of the scope of the current work. We have added a sentence in the text acknowledging this possible limitation:

"Note that the warm air mass seems to be tilted in the vertical but that it is not considered as a Saharan air mass and it does not have direct effects on the surface temperatures. We acknowledge that such a warm upper-troposphere air mass could have indirect effects in surface temperatures due to cloud formation and radiative effects, but analysing this falls out of the scope of the current work"

l 153 and following and Fig 4
I was wondering how similar the clusters between the seasons are. The different colour scales make it difficult to compare by eye, but there seem to be great similarities (referring to the figures in the appendix). I would be interested whether one kind of IT appears in all season, or another exists only in summer, or only in MAM and SON, or something like that. This then raises the question, if similar/the same clusters appear across seasons, is it necessary to split the seasons for the clustering?

*The large-scale structure shows differences across seasons, therefore we decided to keep the results of the clustering separated.* This motivation is explicit in the text and we have changed the color coding of the Figures (5 and S10) to make it more explicit. (ls. 182-184)

"A clustering methodology is employed to make an initial classification of the different intrusions. To account for differences in the radiative forcing, atmospheric circulation and thermodynamics in the Sahara along the year (as seen in Section 3), we apply a clustering analysis of the intrusion events in each meteorological season (DJF, MAM, JJA and SON, henceforth)"

l 168 and following The paragraph starting with "Kendall-tau correlations..." seems to be quite interesting but lacks depth. First, I think there should be a new paragraph, i.e. linebreak. Second, the topic is not introduced at all. It would be helpful to mention the teleconnection indices and then explain that you calculated correlations. Also, you mention you use "some" of the indices but do not motivate which ones. Third, the acronyms used in the equations are not introduced, and the dashes as bullet points can be mistaken for minus signs. Most importantly: I am lacking a discussion of what your resulting numbers mean. I.e. does the first equation imply that IT2 in summer occur more often in a positive AMO phase? Is this reasonable/expected or not? Please elaborate on these results more.

*We agree that the results and discussion around the teleconnection indices is shallow.* Therefore, we have expanded on this and justified the different indices used with the literature of Mediterranean drivers (ls. 221-238):

"We want to take advantage of the separation of intrusion days into different ITs to see if there is any connection between Saharan warm air intrusions and any of the relevant teleconnection indices in the Mediterranean region. There are many large scale drivers that affect the climate of the WMed, some that might be interesting to assess their link with the ITs are: i) The North Atlantic Oscillation (NAO) and it's summer expression (SNAO), which are the leading modes of large-scale atmospheric variability in the North Atlantic and are defined by the pressure gradient between Iceland and Azores (Hurrell, 1995), and through Principal Component Analysis of the summer mean sea-level pressure (Folland et al., 2009), respectively. It has a direct effect on the position and strength of the subpolar jet and therefore affects the circulation, weather patterns and different atmospheric variables in the Euro-Mediterranean (Hurrell, 1995, Folland et al., 2009, Bladé et al., 2012). ii) The Western Mediterranean Oscillation (WMed) is designed to consider the atmospheric dynamics of the Western Mediterranean by taking the pressure gradient between the gulf of Cádiz and the Po plain (Martin-Vide and Lopez-Bustins, 2006). iii) The Arctic Oscillation (AO) (Thompson and Wallace, 1998) is the hemispheric leading mode of variability from surface to stratospheric levels (Dunkerton et al., 1999, Gerber et al., 2010) and it mainly drives the fluctuations of the subpolar jet. iv) Atlantic Multi-decadal Variability (AMV) (Deser et al., 2010) is the low frequency fluctuations of the North Atlantic sea observed in surface and subsurface variables. We use the definition of the Atlantic Multi-decadal Oscillation (AMO) to capture this low frequency variability mode (Kerr et al., 2000, Klotzbach and Gray 2008}, which intends to distinguish itself from the NAO-influenced tripole pattern at interannual time scales (Enfield et al., 2001). It is defined as the detrended average anomalies of sea surface temperatures in the North Atlantic basin (detrending is performed by subtracting the mean global SSTs). The AMO has been linked to Mediterranean temperature fluctuations (Mariotti et al., 2012)."

Also, we discuss the implications of obtaining significant correlations with different teleconnection indices (ls. 242-246).

"The results suggest that a warmer-than-usual North Atlantic could enhance the occurrence of Saharan warm air intrusions in the central and western part of the WMed during JJA; In DJF, having a negative AO (predominance of a low pressure in the northwestern WMed) is correlated with an enhanced number of intrusion events; having a relatively high pressure in the south western part of the WMed with respect to the northeastern part of the WMed (positive WeMO) seems to also favour Saharan warm air intrusions in DJF."

Sect 5
The intrusions are identified based on temperature thresholds, the conditional probability P (heat|intrusion) can therefore be expected to be extremely high, at least in the area that is covered by the affected air mass.

Your approach adds to this as it shows any area that is impacted. However, since the information shown in Fig 7 is a combination of many intrusion days, it cannot be distinguished between areas that are hot "by definition" just because the air mass is present, and areas that are hot on an intrusion day, but not covered by the intrusion itself. Could you please elaborate on this. One idea could be to show Fig 7 for cell-wise calculation of the impact: P (heat in this cell|intrusion in this cell) in order to see how big the difference is between this direct vs your "also -remote" impact.

Another thing about this section is the way you look at the contribution. I spent a long time thinking about this. My immediate question was why don't you show P (heat|no intrusion), as this would be the intuitive measure to compare P (heat|intrusion) to.
I see how your version of the contribution answers the question "In how many percent of heat days was an intrusion present?" and the intuitive conclusion would be "in this many percent of the heat days, an intrusion was present and therefore played a role, i.e. contributed". However, since the heat cannot be the cause of the intrusion, it is difficult to interpret this in terms of causality.
I would definitely ask you to also show P (heat|no intrusion). This shows you how often it was hot, despite no intrusion being present. Contrasting P (heat|intrusion) and P (heat|no intrusion) by looking at a difference, relative risk, odds ratio or log-odds ratio will give insight into how the presence of an intrusion increases or decreases the chance of heat. I.e. when the two probabilities are similar, intrusions do not influence heat very much, while when they are different, you can see the effect of the presence of an intrusion.

We thank very much this thorough comment on the role of the intrusions to extreme temperatures. We have added a contour line showing which is the boundary where intrusion days for each IT were recorded (effectively showing the area of influence, which has been redefined in the text ls. 207-210).
Regarding the metrics used, we have kept the impact as it was in the initial version but we removed the contribution metric. We add the odds ratio as a metric to see which regions have a larger probability of experiencing extreme temperatures in the presence of intrusions.

[Figure]

*Figure R8: Odds ratio of the Impact of the Saharan warm air intrusions on extreme temperatures in the EM region for the different seasons (rows) and ITs (columns). The odds ratio is measured as the fraction of the percentage of intrusion days that coincide with an extreme temperature day (TX90p) divided by the percentage of extreme temperature days that do not coincide with an intrusion day. The area of impact is displayed in green contours, and represents the amount of Saharan air masses recorded in the historical period for a specific IT. The values of the contours go from one Saharan air mass event to the maximum number of days recorded in a gridpoint. Note that the outer contour always represents one recorded day in presence of a Saharan air mass.*

l 259 Here you conclude, that Saharan intrusions in the cold months could be similar to subtropical intrusions. To a reader who is not totally familiar with the literature, it is not obvious why and how Saharan intrusions differ from subtropical intrusions. Please make this more clear.

We want to thank the reviewer for pointing this out. In the first version of the manuscript the concept of subtropical intrusions was poorly introduced and it was difficult to understand what we are trying to convey. We wanted to make a link with the already studied subtropical ridges affecting Europe, as the circulation anomaly composites we obtain are similar to those in the

work by Sousa et al. (2018) and in the visual inspection most intrusions in the cold months look like a subtropical ridge that propagates from the Atlantic. This is also true for some intrusions in the warm months. See below some sentences in the text that attempt to clarify this (ls. 326-328 and 358-364):

"The composites have certain similarities to the composites of subtropical ridges proposed by Sousa et al. (2018). We argue that subtropical ridges might be a main driver of Saharan warm air intrusions, although not a necessary condition."

"…we conclude that intrusions during the cold months could also be mixed with eastern Atlantic air masses at Saharan latitudes. This does not change the fact that Saharan-like air masses are penetrating the WMed, but it is important to note that the provenance can be non-exclusively from the Sahara. Also, from the qualitative assessment of some cold month intrusions (see Figure S4 for an example), we see that cold month intrusions can behave like a broad subtropical ridge with a larger longitudinal extent than warm month episodes. This result is backed by the circulation composites (which will be further discussed in next paragraphs) in DJF, where the upper troposphere positive anomalies are generally broader than in JJA (see Figures 7 and 8); in agreement with the results in Sousa et al. (2018)."

l 273 Here you argue that the increasing number of intrusions can be explained by thermodynamic changes as the warming of the troposphere. I cannot follow that argument. An increase in intrusions according to your detection algorithm could either be due to an increase in the synoptic situations causing the advection, or due to a general warming of the atmosphere in the WMed region. The former is not necessarily connected to thermodynamic changes, but rather to dynamic changes in a warming climate. The latter is closely tied to thermodynamics and a warming troposphere, but it might be a false trend due to the detection method. If your WMed is generally warmer and the latitudinal gradient weaker, your indicator variable thresholds might be exceeded without having advection from the Sahara present. Please elaborate on this in more detail.

We agree with the reviewer, the increase could also be due to a change in the dynamics induced by climate change. We have expanded the discussion of how climate change might also affect changes in the dynamics and in turn have an effect on Saharan warm air intrusions:

"We also observed a statistically significant increase in intrusion days over the historical period for JJA, DJF and SON, compatible with long-term thermodynamic changes and the warming and expansion of the troposphere that can make our climatological definition of Saharan air masses reach higher latitudes more easily or even be generated more to the north. These thermodynamic changes have been extensively linked to global warming (Staten et al., 2018; Webster, 2020; IPCC, 2023), although it is important to consider how climate variability at different timescales can affect temperature increases in the Sahara and WMed regions (IPCC, 2023). Global warming and climate variability could also affect the circulation dynamics (Coumou and Rahmstorf, 2012; Rousi et al., 2022) and directly affect the frequency of mechanisms that lead to Saharan warm air intrusions. This attribution falls out of the scope of

the current work, but we believe such a study should be performed in the future to improve intrusions' predictability."

**Minor Comments**

Throughout the manuscript:

Please be consistent in your spelling and the usage of the terms for the air mass (Saharan warm air intrusion, warm-air intrusion, warm intrusion, intrusion, air mass, etc.). It will help the readability if you use one clear term throughout the manuscript. Examples, e.g. 50,53,57 or the title of Section 5. Thanks for the suggestion, we refer to them as Saharan warm air intrusions in the text, and we often use simply "intrusion days" for shortness when it is clear that we refer to Saharan warm air intrusions.

Similarly: Use cross-section (or a precise term of your choosing), as "cut" can be misunderstood (you use cut, vertical section,...). We changed all instances to cross-section

l 14 You are referring to changes over what time period? Climatological short(er) term changes? *We refer to both historical and projected.* We have changed it to "Numerous studies discussing the Mediterranean climate have underscored important historical and projected changes in temperature, precipitation, and their extremes"

l 38 this should be "forest fires in Portugal" It has been changed in the text

l 45 gain insight into it has been modified

l 46 applicable It has been changed

l 50 I don't understand this sentence. These are the aims of the study, no? What do you mean by "objectives to characterise intrusions into the WMed"? Sentence is changed to "This study has four objectives with regard to Saharan warm air intrusions into the Western Mediterranean …."

l 65-66 This was already mentioned very similarly in l. 43 and following. The last sentence has been removed.

l51/69 Please define the "historical period" in terms of years somewhere. It has been added in "during the recent historical period (1959-2022)"

l 69 Do you mean daily resolution? Yes, thanks. It has been changed to daily resolution

l 73 Daily maximum temperature at what level? It is surface level and has been added to the revised text

l 74 Please define the WMed and Euro-Mediterranean region in terms of lon/lat. It has been added "...intrusion events in WMed (10ºW, 20ºE, 35ºN, 44ºN)..."

l 77 For better readability I suggest:

"We adapt the definition of Sousa et al. (2019), which takes into account..."

We have changed the text according to the reviewer's suggestion

l 81-82 Please reference equations 1 and 2 They have been referenced in the preceding paragraphs where the variables are defined

l 90 the gas constant of dry air It has been modified

l 106 "the detected events" Thanks, It has been modified in the text

l 120 Why are you showing August 2006 here and May 2015 in Fig 1? Are they specifically representative? But why chose 2 different ones? *There is no specific reason for choosing one or the other. Both correspond to intrusion days. We think it is nice to show different events.*

l 121 Sardinia is not at 17◦E? Do you mean the air mass penetrates east of Sardinia? also: "intense trough in the North Atlantic". *That is right.* I meant Sicily, it has been corrected in the text

l 132 "different expressions" is not a precise term, maybe rather use "properties"? It has been changed to "properties" in the text

l 137 I am not sure whether I understand this clustering correctly. You use ∆GH and θ in the WMed region on days that were identified as intrusion days? And then you want to cluster those days into different types. Your results seem as if the position of the anomalously high indicator values defines the clusters. But would the clustering also pick up on two clusters that affect the same region but with different magnitude? *From the obtained results we gave more weight to the interpretation of the positioning of the intrusion, but it is true that the clustering seems to also capture differences in the magnitude (IT2 of Figure 4 clearly has larger magnitude as well as other ITs in the rest of seasons). Thanks for the comment, we will include these findings in the results and discussion.* The changes have been added to sections 4, 5, 6 and 7. An example: "... the ITs can be distinguished by the location where the anomalously warm air mass is found and in its intensity (how big the anomaly magnitudes are)". This has broadened the discussion about intrusion types.

l 151 Looking at the plot for JJA, I cannot really see how you conclude on 3 clusters. The elbow seems to be higher and the high value in the Silhouette score looks a bit like an irregularity? We consider the peak in the Silhouette score as a relative maxima although the pseudo-f seems to have the elbow at k=4. We acknowledge that the choice of N is rather arbitrary, but we prioritise a consistent criteria across seasons, and therefore we take into account the information from both the Pseudo-f and Silhouette scores.

l 151 The last sentence in this paragraph "Finally,..." is really inherent to the clustering method and not necessary here. We prefer to keep the sentence to make it explicit in case some readers are not familiarized.

l 155 You are referring to Fig S6, not 5 here. Thanks for pointing this out. In the revised text is Figure S9 now

l 162 I think this should be DJF and SON, you are referring to panels a,c. *It is DJF, JJA and SON which are panels a,c,d. It has been fixed.*

Fig 6 How do you handle the duration of events, where a long period of intrusion is interrupted by a very short period, where the 5% criterion is just not met for a short time? If you handle these as separate events, this would reduce the mean duration very much compared to when you use a filling or filter. There are 5 instances where there is one day between events, and 7 instances where 2 days separate two events (this represents less than 1% of the amount of intrusion event onsets), and therefore we estimate it is not playing a role in the final results. Nevertheless, If we think about the meaning of these one or two days in between intrusion days, it can be either because the air mass from the same event retreated and went back inside the WMed, or because there were two successive events. If we are in the latter case but we consider the events to be the same we will be erroneously merging two events. Contrarily, If we are in a situation where the same event retreated just for one or two days and count it as two

separate events we will be counting an extra event and reducing the mean event duration. Having acknowledged that, we think it is better to commit the second error type because the mechanism that makes a decaying event turn back into an intrusion event could be considered as a "trigger mechanism" as the event would end without it.

l 167 from Fig S8, to me it looks more like IT1 and IT2 often transition to IT3, not IT1 to IT2 and IT3? Or are you interpreting 3 in more than 20 as often? *That is right, IT1 and IT2 often transition to IT3.* It has been addressed in the text: "ITs are generally persistent in JJA, although IT1 and IT2 can often transition to IT3."

l 178 This is not the first time you use the term Euro-Mediterranean. Please introduce the acronym and the exact definition when you mention it first (in the data section?) It has been added in the data section: "...in the Euro-Mediterranean region (roughly 20ºW, 50ºE, 30ºN, 70ºN)."

l 180 the 90th percentile of what variable (there is simply a word missing I think). It has been addressed: "We compute extreme temperature days for the historical period at each grid cell, estimated as those days that exceed their daily maximum surface temperature 7-day rolling window climatological 90th percentile."

l 197 What do you mean with "largest contributions come from JJA"? *The largest contribution of extreme temperatures that come from intrusion days.* Meaning that in JJA the percentage of intrusions contributing to extreme temperatures is the highest. We are not showing this metric anymore but we mention its values (the percentage of TX90 days that coincide with an intrusion day): "...if we compute, for every season, the percentage of TX90p days that coincide with an IT*n* intrusion day we see that in some regions intrusions contribute up to ~10% of the seasonal TX90p days in JJA and up to 4% in DJF (not shown)."

l 212 subtracting them from the intrusion days' composite It has been addressed.

l 217 "for" instead of "from" in the parentheses It has been addressed.

l 223 What do you mean with "the [...] circulation and SLP anomalies are coherent"? It has been made clearer in the text: "...as the wind anomalies follow an (anti-) clockwise pattern around anomalous (Highs) Lows."

l 224 What is the tilde over the 3? *It is a typographic error, thanks for noticing.*

l 225 always It has been corrected

l 228 negative sea level pressure anomaly "anomaly" has been added

l 243 Your description earlier implied you are detecting intrusions that originate in the entire Sahara, why are you mentioning only the western Sahara here now? *This is a typing error.* It has been modified to only "Sahara"

l 251 You do not really show that the threshold you choose is robust, or how sensitive result are to it's choice. *We will avoid saying robust.*

l 256 What is a "climatological air mass"? I understand what you are referring to, but this can be misleading. Also "the latitudinal band" is not precise. We reworked the text in line with the second major comment. This sentence has been changed to:
"During the cold months, the climatological values of the indicators (...) over the Sahara do not seem exclusive to the Sahara, but rather have similar properties in regions outside the Sahara at the same latitude (~ 15ºN-35ºN,30ºW-50ºE)"

l 262 Please leave out the "backward", as the studies you cite use forward and backward trajectories. It has been addressed

What do you mean by "historical contribution"? It has been changed to: "The contribution of Saharan warm air intrusions in the WMed during the historical period is important."

l 292 remove parenthesis, "which" instead of "and" afterwards It has been addressed

**Comments about Figures**

While many of your figures are nice in general, all have very small font sizes on the axis labels, the axis ticks, etc. Please change this for better readability. Often the use of shared axes or removal of figure titles that are redundant with the caption can create space. Thanks for the suggestion, we have enhanced font sizes overall.

MAJOR: Fig S1 Caption is wrong, the red box is not the Saharan region. Also here the figure title is redundant. Axis ticks are not necessary in all rows and columns but only at the bottom and left, which makes space for bigger labels ad better readability. Please mention what contours and shading represent in the caption.

Thanks for pointing this out, it has been modified in the new version of the figure and caption.

Fig 1: The green and cyan lines are not so easily distinguishable, please use additional different linestyle. The black dashed box is supposed to be an example for how big 5% are, this needs to be stated very clearly, otherwise it is not obvious that this is just exemplary.

We have changed the line colors and improved the explanation of the 5% in the text and in the figure caption.

Fig 2 I do not see the additional value of this figure. When it is mentioned in line 96 it is rather confus- ing. As far as I understand the helpful information here is that the averages over the Saharan region are in the high tails of the distribution of the WMed region, I think this can be mentioned without showing the figure. One other question this figure raised for me is the spread in the blue dots in February.

We have moved this figure to the supplement and we now support the choice for the indicators with information from Figures R1 and R2 (Figures S1 and S2 in the revised Supplementary material).

Fig 3 Add dates to the caption when the exact time of the event is, and what you refer to as before, dur- ing and after.

We have improved the caption and added the dates of the event and those of the days before and after the event

Fig 4 Caption could be clearer. These are composites of the intrusion days of the respective type in the JJA season? The labels in the contour lines are difficult to read and interpret.

We have improved the caption

Fig 5 As IT 1-3 are categorical values, a non-sequential colourmap would make the perception easier. Espe- cially in connection with the figures in the Supplement, where more ITs exist. Then the question is also, relating to an earlier comment of mine, whether all IT1 across seasons are similar to each other or not - using the same colour for all IT1 implies that they are more similar to each other than to others.

We have changed the color code for the different ITs so that is clear that they are distinct.

Fig 6 Do I understand it correctly, that the orange bar can never be larger than the blue? Consequently, when I only see orange, this means it is a single event?

*Exactly, when only orange appears means that only one event exists for that season and year.* We now mention this in the caption.

Fig 7/8/9 Can you add the area of influence as a contour. You discuss the relation between the impact and the area of influence (e.g. l 192), but it is difficult to see that by comparing differnt plots on different pages.

We have adde the contours of the area of influence in green.

Fig 9 be constistent in use of SLP and PSL, I think it should be wind speed anomalies in the caption here. And what arrows are you referring to in panels (a) that are not the quivers? This caption is not easy to follow. It might also help to add numbers to the climatological contours for reference. And the quivers are way too small to see their direction properly.

We have changed every instance to SLP.

Is Fig 9 panel (a) the same as Fig (4) panel (b) (except for the significance masking)???

*Fig 9 (a) and Fig 4 (b) are not the same. Any similarities are due to compositing the same days, although the variables are different*

REVIEWER #2 COMMENTS

**General Comments:**

This manuscript is devoted to studying Saharan warm air intrusions in the Western Mediterranean during a historical period (1959-2022), emphasising not only their identification methodology (considering different intrusion types according to their location), and the resulting creation of a catalogue of events, but also the assessment of their impacts on air temperature extremes and the analysis of the large-scale atmospheric driving mechanisms, namely upper-tropospheric anomalies over North Africa. Overall, the topic of research is within the scope of WCD and pertinent under a climate change context and increasing occurrence of temperature extreme events. The datasets used (daily tropospheric variables obtained from the ERA5 reanalysis) are adequate for the purposes of the study, whereas the methodology to identify the different intrusion types provides some novelty. The methodological approach for the detection of Sahara air intrusions in Western Europe is simple and effective, with low computational costs. The results are scientifically sounding, based on sufficient evidence provided by the authors and are in line with previous research outcomes. Generally, the figures are of good quality and the length of the manuscript is adequate. Therefore, I recommend the publication of the manuscript after some revisions that are outlined below in the specific comments.

We want to thank the reviewer for taking the time to read the text and provide feedback that will greatly improve the quality of the text

**Specific Comments:**

Abstract: You mention four ITs. However, depending on the season, different numbers are obtained and considered in the manuscript. Please revise.

*Thanks for pointing this out.* The number of ITs per season is now specified in the abstract.

Section 1: There is some unnecessary repetition of citations to previous studies (e.g., Sousa et al. 2019), while other relevant references are still missing (e.g., 10.3402/tellusa.v67.26032, among others). I suggest a more in-depth literature review, as many previous studies have already linked temperature extremes in the Western Mediterranean to atmospheric large-scale mechanisms and anomalies. This is particularly pertinent to corroborate the present study results and provide a good state-of-the-art on this topic of research, giving sufficient credit to other preceding studies.

Thanks for pointing this out. The literature on the link between temperature extremes in the Western Mediterranean and atmospheric large-scale mechanisms and anomalies has been expanded in the introduction (ls. 33-37 and 42-47):

"In the broader Euro-Mediterranean region several studies have used different approaches to better understand the provenance and evolution of air masses that end up producing extreme temperatures at certain locations, through case studies (Rous et al., 2023, Hotz et al., 2024, Mayer and Wirth, 2025} and analysing extreme temperature events catalogues (Santos et al., 2015, Zschenderlein et al., 2019). While some of the results hint that there is air advected from the Sahara in some of the events, there is no specific study that attempts at generalising the effects of Saharan warm air intrusions."

(...)

"Several works have hinted at the impact that southerly flow from Africa might play a role in Euro-Mediterranean temperatures. (Sousa et al., 2019) studied the impact that past intrusion events have on heat waves in the Iberian Peninsula, Pereira et al. (2005) demonstrated the effect that advections from northern Africa can have on forest fires in Portugal, Santos et al. (2015) suggested that an anticyclonic circulation over the Northwestern Africa generates extremely high temperatures in the Iberian Peninsula, and Zschenderlein et al. (2019) show how Air masses in africa can be linked to extreme temperatures in Central Mediterranean; among others (Sousa et al., 2018, Rousi et al., 2023)"

Ln 50-65: I would avoid bullet-point lists. Providing more compact text is commonly preferable in scientific manuscripts.

*Thanks for the suggestion. We considered bullet-point lists to make the objectives and structure of the work more explicit.* We have kept the objectives as a list to make them more visually available, and we have merged the sections into the text as the reviewer suggests.

Ln 54: please avoid using "etc". Be more specific.

*Thanks for pointing this out,* it has been changed to "spatial distribution, seasonality and trends"

Section 2: The dataset selected for the analysis was ERA5. Although this choice is adequate for the goals of the study, it should be complemented with weather station data (e.g., from ECA&D),

as reanalyses commonly present important biases in near-surface weather conditions, particularly in capturing small-scale spatial variability and local weather/climate patterns. The use of maximum temperatures (TX) recorded at local weather stations would help validate the occurrence of temperature extremes. Hence, I suggest the incorporation of some weather station data to confirm and better quantify the intensity of the extreme events. From my point of view, this is an important limitation on the robustness of the study, though it is not expected a major impact on the results.

We want to thank the reviewer for their comment as adding some observational results will make the study more robust. Therefore, we have included results from some ECA&D stations in the Euro-Mediterranean region. As depicted in the reworked Figure 6 (corresponding to the initial submission Figure 7). A table in the supplementary material has been added to document the stations used.

[Figure]

Figure 6: Impact of the Saharan warm air intrusions on extreme temperatures in the EM region for the different seasons (rows) and ITs (columns). The impact is measured as the percentage of intrusion days that coincide with an extreme temperature day (TX90p). The amount of intrusion days for each season and IT is specified in the title of each panel. Colored dots are results from

ECA&D station data. The area of influence is displayed in green contours, and represents the amount of Saharan air masses recorded in the historical period for a specific IT.

Section 3:

- Please do not use physical units in italic font. it has been addressed in the revised text.

- Rd is the "dry air gas constant". *Thanks for pointing this out,* it has been addressed in the revised text.

- Cp should be cp (lowercase C). *Thanks for the suggestion,* the text has been modified.

Figure 1: the dashed black box within the Western Mediterranean box is not sufficiently explained in both the caption and text. It seems that the selected day for Fig. 1 does not fulfil the corresponding area requirement.

*Thanks for pointing this out.* This point has been explained in a better way in the revised text (and in the figure caption): e.g. "The dashed black box in Figure 1 is a representation of how much area is 5% of the WMed, but note that it can take any shape."

Ln 101: "rolling" or "moving"?

*In the text we use "rolling" and "moving" (means or windows) interchangeably.* We have homogenised all instances to "rolling" to be coherent.

Ln 103 & 104: "red shaped" or "orange"? It has been addressed: "the area is painted orange to represent a Saharan air mass"

Figure 2 shows a strong positive correlation between both variables. The same is clear in Figure 4. This hints at the high level of redundancy (colinearity) between variables and the information they bring to the analysis. This point should be better discussed, namely its potential implications on the results and the reasoning for their use under the occurrence of temperature extremes driven by Saharan warm air intrusions. For instance, why are you using two variables that deliver similar information and not using variables that complement each other?

We want to thank the reviewer for pointing this out. *The two variables are correlated but not perfectly, and therefore they can complement each other. Using the geopotential thickness and the mean potential temperature aims at identifying air masses with a sufficiently high temperature, and a consistent air mass volume.* Furthermore, the average potential temperature is conserved during adiabatic heat exchange and therefore is a good complement to changes in the geopotential thickness. We added this to the revised text (ls. 89-92):

"Two indicators can inform us about these air mass characteristics: the geopotential thickness between levels 1000 and 500 hPa ($\Delta GH_{500-1000}$, henceforth) defined in Equation 1 and the average potential temperature between levels 925 and 700 hPa ($\theta_{700-925}$) defined in Equation

2. Note that the latter is conserved in adiabatic heat exchanges and therefore is a good complement to the former.

Figure 3: the colour scales are not adequate for representing potential temperature (I suggest a rainbow scale or similar). Please revise. Further, colours are not consistent between panels and the description in the caption.

*Thanks for pointing this out, the problem we find with the rainbow map is that it is not suited for certain types of colorblindness. We will maintain the Colormaps with python's "inferno" (https://matplotlib.org/stable/gallery/color/colormap_reference.html) coloring if there are no major objections.*

[Figure]

Ln 140: please remove ", similar to". It has been addressed in the revised text.

Section 7: I suggest splitting the Discussion and Conclusions section into two separate sections.

*Due to the nature of the current work the authors believe the Conclusions and Discussions are more suited together than split into two sections.*

Ln 275: IPCC in uppercase. It has been addressed

Ln 292: "as in (Sicard et al., 2022)". Please revise the citation formats throughout the manuscript. *Thanks for the suggestion.* They have been addressed in the revised text.

Last paragraph of Section 7: the possible linkage of the upper-tropospheric anomalies underlying the onset and establishment of Saharan intrusions to the westerly eddy-driven jet stream over the North Atlantic, including anticyclonic/cyclonic wave-breaking processes, should be discussed.

*Thanks for the suggestion.* In the revised text we have added the need to understand possible links between the upper-troposphere anomalies and the North Atlantic and eddy driven jet that might trigger/drive Saharan intrusions onset. While there is still work to do, we think it falls out of the scope of the current study. Therefore, we propose this link and cite relevant literature that has assessed similar topics (10.1175/JCLI-D-20-0658.1, https://onlinelibrary.wiley.com/doi/10.1002/qj.625) to open up further wo

---

## Referee Report (RR1)

**Review of: "Saharan warm air intrusions in the Western Mediterranean: identification, impacts on temperature extremes and large-scale mechanisms" by Cos et al. - revised version**

I thank the authors for answering to my comments in depth and improving the manuscript. The manuscript has improved a lot with the current version, however, I still have some minor concerns that need to be addressed before publication, most of which are technical, but some more substantial.

**Minor comments**

**line 72** I think if you explain all your sections like this, you should indeed mention all of them. You do not mention Section 7 here.

**line 81** What do you mean by roughly? You must somehow have a specific region that you use, i.e. to compute the TX90 values? But thank you for clarifying.

**line 89** double parentheses could be avoided

**line 110, Fig S3** In the caption you refer to a red box, which in fact is black

**line 124, Fig S4** y-axis and caption differ. Is it $\bar{\theta}_{700-925}$ or $\bar{\theta}_{750-925}$?

**line 133,135** be consistent with red vs orange shading

**line 132, Fig 1** be consistent with purple/fuchsia

**line 135** "...a representation of how much area 5% of the WMed IS, ..."

**line 139-141** Thank you for adding the sensitivity study. I thought this result was really interesting. My immediate reaction to this sentence was 'why?'. Maybe you could add a thought about what mechanism leads to these changes in the length of the detected events depending on the lower level bounds.

**line 158** in panels a-d you show seasonal results, not monthly as you state in this line, or did I misunderstand?

**line 159-160** And Fig2: "Results show that the months with most of the intrusions are July and August, as well as December. There is also some notable activity during January-February and May-June." Thank you for adding the panels e-h) in Fig2. However, I still have some reservations about this.
First: panel indicators are missing in the figure.
Second: panel a has a different y-scale, which makes it very difficult to compare the panels and follow your reasoning.
Third: The boxplots are hard to read as they are small, the circles big, and the y-axis too tall, so that it is squished in the bottom. Also, the median is marked in orange, which can be confusing as the data you show in the box plots refers to the blue bars not the orange ones in panels a-d.
Apart from these technical/visual things, I cannot follow the sentence I pasted above. To me it does not become obvious that e.g. December is has many intrusions, only that the spread seems higher? Also you say Jan and Feb have some activity, but from the plot they don't look too different from March or the Autumn months. This definitely needs to be addressed and made more clear before publication.

**lines 213-216** These 2 sentences do not belong in the discussion about the ITs, but rather right after line 160.

**line 229** Are you using the same acronym for WMed region and WMed Oscillation? This is confusing.

**line 233** double parentheses. Also 'is' and 'fluctuations' - plural or singular?

**line 246** I still think that the itemised list of correlations should not be itemised using dashes, as they can be read as minus signs

**line 255** double parenthesis should be avoided

line 257   There is no table S2, it should be S1, and then in the supplementary you should call it that as well, it is 'Table 1' at the moment

"We aim to quantify the probability of getting..."

lines 265-267   This finding is really interesting. However, you do not really explain why you think this seasonal difference exists.

lines 270 ff   'impact on'

I was wondering here: S12 looks similar for all seasons. But you motivated it in order to explain what is happening in JJA. I am missing your reasoning then, if the behaviour in JJA is explained by a slower dynamics, or what else it could be.

line 280   Thank you for following my suggestion on how to look at the conditional probabilities. However, the equation you show here is not the odds ratio, but the relative probability or relative risk. While it has a similar meaning, please be careful about the wording and the exact conclusions you draw. Decide for either of them, but then check the meaning and be consistent.

For reference, the odds ratio is defined as:

$$O1 \quad = \quad \frac{P1}{1 - P1} \tag{1}$$

$$O2 \quad = \quad \frac{P2}{1 - P2} \tag{2}$$

$$OR \quad = \quad \frac{O1}{O2} \tag{3}$$

where in your case: $P1 = P(TX90p|ITnday)$, $P2 = P(TX90P|noITnday)$

line 281   '...when the day is not in ITn' A day cannot be IN an intrusion type? please rephrase.

line 283   "isn't" should be is not

line 289   to have an influence ON

line 292   the word coincide indicates a joint probability, not a conditional one

Fig 7   Thank you very much for showing this, it improves the manuscript greatly. I had a follow-up question now. Did you look at the same thing for all ITs combined? I.e. $P(TX90p|intrusionday)/P(TX90p|nointrusionday)$. Since the intrusion types are still somewhat similar, comparing to "no ITn intrusion day" will still incorporate all other ITs (which have an effect on similar regions). I assume you could see even more clearly that intrusions in general have an effect on temperature if you showed it for intrusion days in general.

line 306   you are referring to Fig4 not 5 here?

line 328   change the passive to an active citation

Fig 8/9   Your description is a little bit hard to follow, since the colour scales are different, which makes it difficult to estimate if a signal is stronger in Fig 8 or 9.

line 351   reference Fig 7 here

line 363   S4 is the wrong figure to reference here (since this is not the only instance of a wrong figure being referenced, please check carefully if all references are correct)

general   be consistent in your spelling of upper-troposphere/tropospheric, i.e. with or without hyphen

---

## Author Response (AR2)

We would like to thank the reviewer for the very thorough review. We hope all small technical errors are now solved and we believe that the questions and suggestions provided in this round of revisions make the study more robust.

I thank the authors for answering to my comments in depth and improving the manuscript. The manuscript has
improved a lot with the current version, however, I still have some minor concerns that need to be addressed
before publication, most of which are technical, but some more substantial.

**Minor comments**

line 72 I think if you explain all your sections like this, you should indeed mention all of them. You do not mention Section 7 here.

Thanks for the suggestion, section 7 has been added in the manuscript structure.

line 81 What do you mean by roughly? You must somehow have a specific region that you use, i.e. to compute the TX90 values? But thank you for clarifying.

TX90 values are calculated at specific grid points (see "...we compute extreme temperature days for the historical period at each grid cell…" in the manuscript), so there is no issue in taking a larger or smaller domain and results will remain the same. I agree, though, that the "roughly" is a bit ambiguous, so I'll remove it as we are using that exact domain.

line 89 double parentheses could be avoided

Thanks for noticing it has been addressed.

line 110, Fig S3 In the caption you refer to a red box, which in fact is black

Thanks for noticing this error, the caption has been fixed.

line 124, Fig S4 y-axis and caption differ. Is it θ700−925 or θ750−925?

It is 700-925 as in the caption, the figure has been fixed. Thanks.

line 133,135 be consistent with red vs orange shading

Thanks for noticing the discrepancy, I changed it to orange.

line 132, Fig 1 be consistent with purple/fuchsia

It has been changed to fuchsia.

line 135 "...a representation of how much area 5% of the WMed IS, ..."

Thanks for pointing this out, it has been fixed.

line 139-141 Thank you for adding the sensitivity study. I thought this result was really interesting. My immediate reaction to this sentence was 'why?'. Maybe you could add a thought about what mechanism leads to these changes in the length of the detected events depending on the lower level bounds.

We agree that this is an interesting point, thanks for bringing it up. We think we can hypothesise about that only for the geopotential height, where changes are slightly more noticeable. Therefore, we suggest that obtaining longer and more intrusion events when using an upper lower-bound of the potential temperature metric this is due to the fact that separating the indicator's lower bound from the ground diminishes diabatic cooling with the surface and therefore the Saharan air masses will change slower when taking potential temperature at 850 hPa. We modify the text about sensitivities to:

 "In general, using a higher lower-level potential temperature bound leads to more and lengthier events, and a higher lower-level geopotential bound leads to a very slight decrease (increase in JJA) in the amount and length of the events. The increase in events due to taking 850 hPa potential temperature might be related to the diminished diabatic cooling of the air mass as the lower bound of the indicator is further away from the surface. Changes due to GH lower bound are less obvious and we can't provide a robust hypothesis for their small sensitivity."

line 158 in panels a-d you show seasonal results, not monthly as you state in this line, or did I misunderstand?

I agree, it has been changed to seasonal. Thanks for pointing this out.

line 159-160 And Fig2: "Results show that the months with most of the intrusions are July and August, as well as December. There is also some notable activity during January-February and May-June." Thank you for adding the panels e-h) in Fig2. However, I still have some reservations about this.
First: panel indicators are missing in the figure.
Indicators have been added, thanks for pointing it out.
Second: panel a has a different y-scale, which makes it very difficult to compare the panels and follow your reasoning.
The y-scale was actually the same but the 10 and 30 ticks were missing. It has been fixed.
Third: The boxplots are hard to read as they are small, the circles big, and the y-axis too tall, so that it
is squished in the bottom. Also, the median is marked in orange, which can be confusing as the data you
show in the box plots refers to the blue bars not the orange ones in panels a-d.

The boxplots have been changed to violin plots spanning the whole distribution. We think it is now easier to compare the different months.

Apart from these technical/visual things, I cannot follow the sentence I pasted above. To me it does not

become obvious that e.g. December is has many intrusions, only that the spread seems higher? Also you

say Jan and Feb have some activity, but from the plot they don't look too different from March or the

Autumn months. This definitely needs to be addressed and made more clear before publication.

We understand the concern. After producing the violin plots it is now easier to see that the distributions of January, February, March, May and June look quite similar and some intrusion activity is present, although saying "notable activity" might be a bit much. Therefore we changed the sentence to:
"There is also some activity during January-March and May-June."

lines 213-216 These 2 sentences do not belong in the discussion about the ITs, but rather right after line 160.

I agree, thanks. They have been moved.

line 229 Are you using the same acronym for WMed region and WMed Oscillation? This is confusing.

Wrong acronym, thanks for noticing. The acronym should be WeMO and it has been changed in the text.

line 233 double parentheses. Also 'is' and 'fluctuations' - plural or singular?

The double parenthesis has been fixed and changes "fluctuations" to "fluctuation".

line 246 I still think that the itemised list of correlations should not be itemised using dashes, as they can be read as minus signs

True, the dashes have been removed. Thanks.

line 255 double parenthesis should be avoided

It has been addressed in the text.

line 257 There is no table S2, it should be S1, and then in the supplementary you should call it that as well, it is
'Table 1' at the moment

Thanks, it has all been homogenised to Table S1.

"We aim to quantify the probability of getting..."

Thanks for the suggestion, it has been modified in the text.

lines 270 ff 'impact on'

It has been addressed, thank you.

lines 265-267 This finding is really interesting. However, you do not really explain why you think this seasonal difference exists.
I was wondering here: S12 looks similar for all seasons. But you motivated it in order to explain what is happening in JJA. I am missing your reasoning then, if the behaviour in JJA is explained by a slower dynamics, or what else it could be

True, what we can say from S12 is that the reason why the impact beyond the area of influence is very low in JJA is not because the circulation is slower. The impact in the day after an event does not increase. We add this to the text:
"Therefore, Figure S12 disproves that JJA intrusions might move slower and have an impact outside the area of influence the days after an intrusion event. This result suggests that if JJA has no impact outside the area of influence it is because Saharan warm air intrusions, and their impacts on extreme temperatures, are simply more confined to the area of influence.".

line 280 Thank you for following my suggestion on how to look at the conditional probabilities. However, the equation you show here is not the odds ratio, but the relative probability or relative risk. While it has a similar meaning, please be careful about the wording and the exact conclusions you draw. Decide for either of them, but then check the meaning and be consistent. For reference, the odds ratio is defined as:
$O1 = P_1 / 1 - P_1$ (1)
$O2 = P_2 / 1 - P_2$ (2)
$OR = O1 / O2$ (3)
where in your case: $P_1 = P(T_{X90p}|IT_{nday})$, $P_2 = P(T_{X90P}|noIT_{nday})$

We want to thank the reviewer for pointing this out, we agree that the metric we showed in the revised text is not the odds ratio but rather the risk ratio. See Figure R1 for the results of the Odds ratio.

The two metrics do not have exactly the same meaning, but the conclusions we arrive at are the same. For simplicity we will keep the risk ratio metric, so we adapt the text to:

"To quantify how the presence of an intrusion increases or decreases the risk of having TX90p days we compute the risk ratio (Equation 4) between the probability of having TX90p

conditioned on having an ITn intrusion day versus that of having TX90p conditioned on being on any other day than the ITn intrusion day.

(…)

The risk ratio is the ratio of how much more risk there is to have an extreme temperature event if the day is an ITn intrusion day than if it is not. Risk ratio will be closer to one when the two probabilities are similar, and therefore the effect of the intrusion is not affecting the probability of having a TX90p day. When the risk ratio has values above 1 it means that having an extreme temperature is more probable when an intrusion is present. Risk ratio under one means that having an extreme temperature day is more probable on non-intrusion days. Values above 10 suggest a strong difference in the occurrence of TX90p between having or not an ITn intrusion day (Ellison 1996).

Figure 7 displays the risk ratio and we see that …"

[Figure]

Figure R1: Odds ratio of days with and without Saharan warm air intrusions to the extreme temperature days (TX90p) of each season (rows) and IT (columns). It is computed as the fraction of the impact divided by the probability of having an extreme temperature day when no intrusion is recorded. Colored dots are results from ECA&D station data (Table S1). The area of influence is displayed with a green (dotted) contour, and represents the limit where Saharan air masses have been recorded in the historical period for a specific IT.

line 281 '…when the day is not in ITn' A day cannot be IN an intrusion type? please rephrase.

The text has been changed to: "when the day is not an ITn day".

line 283 "isn't" should be is not

The text has been addressed, thanks.

line 289 to have an influence ON

Thanks for pointing this out, it has been fixed.

line 292 the word coincide indicates a joint probability, not a conditional one

The text has been addressed: "the percentage of TX90p days that are also an ITn intrusion day".

Fig 7 Thank you very much for showing this, it improves the manuscript greatly. I had a follow-up question now. Did you look at the same thing for all ITs combined? I.e. P (T X90p|intrusionday)/P (T X90p|nointrusionday). Since the intrusion types are still somewhat similar, comparing to "no ITn intrusion day" will still incorporate all other ITs (which have an effect on similar regions). I assume you could see even more clearly that intrusions in general have an effect on temperature if you showed it for intrusion days in general.

Thanks for the suggestion. We performed a study of the impact on extreme temperatures of all the intrusion days in each season at the beginning of this work. We saw that the impacts were smoothed over the EM region due to the spatial heterogeneity of the Sharan warm air intrusions. That is why we use ITs to help us discriminate between the impacts in the different parts of the WMed.
It is true that in the risk ratio there are intrusion days being included in the no ITn days. Nonetheless, we think the effect of those days is minimal and actually is useful to include them in the no intrusion days: For example, the impact of IT1s in the east of WMed is generally very small (and the risk ratio is sometimes below 1). Therefore, counting that IT1 intrusion day decreases the influence of intrusions to temperature extremes in the east. This is why we want to keep the ITs separated.
Below we show Figure R2 where we compute the relative risk or risk ratio for all season's intrusion days and it can be seen how, even if the risk ratio is positive in most of the EM region (and especially strong in DJF and MAM), the temperature effects decrease in magnitude for the reasons explained above (see how the impacts in eastern Mediterranean are very reduced).

[Figure]

Figure R2: Risk ratio of having TX90p days with and without Saharan warm air intrusions for each season (rows). It is computed as the fraction of the impact divided by the probability of having an extreme temperature day when no intrusion is recorded

line 306 you are referring to Fig4 not 5 here?

True, thanks for pointing this out.

line 328 change the passive to an active citation

I have applied this change in the text. Thanks for noticing.

Fig 8/9 Your description is a little bit hard to follow, since the colour scales are different, which makes it difficult to estimate if a signal is stronger in Fig 8 or 9.

I agree, thanks for the suggestion. We changed the color bars and quivers to the same scale for all seasons. The differences between seasons should now be clearer.

line 351 reference Fig 7 here

It has been added in the text.

line 363 S4 is the wrong figure to reference here (since this is not the only instance of a wrong figure being referenced, please check carefully if all references are correct) general be consistent in your spelling of upper-troposphere/tropospheric, i.e. with or without hyphen

It has been addressed. Thanks for pointing out the issues with the figures, we have gone over all references to make sure that they are correct.